# Pathway-Specific Genomic Alterations in Pancreatic Cancer Across Populations at Risk

**DOI:** 10.3390/ijms26167695

**Published:** 2025-08-08

**Authors:** Cecilia Monge, Brigette Waldrup, Francisco G. Carranza, Sophia Manjarrez, Enrique Velazquez-Villarreal

**Affiliations:** 1Center for Cancer Research, National Cancer Institute, Bethesda, MD 20892, USA; 2Department of Integrative Translational Sciences, Beckman Research Institute, City of Hope, Duarte, CA 91010, USA; 3City of Hope Comprehensive Cancer Center, Duarte, CA 91010, USA

**Keywords:** pancreatic cancer, targeted therapy, cancer disparities, genetic mutations, precision medicine, TP53 pathway, WNT pathway, PI3K pathway, TGF-Beta pathway, RTK/RAS pathway

## Abstract

Pancreatic cancer (PC) is a highly aggressive malignancy with increasing incidence and poor survival. Hispanic/Latino (H/L) patients, despite having a lower overall incidence than Non-Hispanic White (NHW) patients, are often diagnosed younger and at more advanced stages, leading to worse outcomes. The molecular mechanisms underlying these disparities remain unclear. This study characterizes mutations in key oncogenic pathways—TP53, WNT, PI3K, TGF-Beta, and RTK/RAS—among H/L and NHW patients using publicly available datasets. We analyzed genomic data from 4248 PC patients (407 H/L; 3841 NHW), comparing mutation frequencies across pathways. Chi-squared tests assessed group differences, and Kaplan–Meier analysis evaluated survival outcomes by pathway alterations. TGF-Beta pathway mutations were less common in H/L patients (18.4% vs. 24.4%, *p* = 8.6 × 10^−3^), with notable differences in *SMAD2* (1.5% vs. 0.4%, *p* = 6.3 × 10^−3^) and *SMAD4* (15% vs. 19.9%, *p* = 0.02). While overall differences in other pathways were not statistically significant, several genes showed borderline significance, including *ERBB4, ALK, HRAS, RIT1* (RTK/RAS), and *CTNNB1* (WNT). No significant survival differences were observed in H/L patients, but NHW patients with TP53 alterations showed borderline survival associations. This study reveals ethnicity-specific pathway alterations in PC, with *SMAD2, ERBB4, ALK,* and *CTNNB1* mutations being more frequent in H/L patients, while *SMAD4* and *PI3K* alterations had prognostic value in NHW patients. These findings indicate the importance of incorporating ethnicity-specific molecular profiling into precision oncology for PC.

## 1. Introduction

Pancreatic cancer (PC) is an aggressive malignancy and one of the leading causes of cancer-related mortality worldwide, with incidence rates continuing to rise [1,2]. Despite advances in treatment, PC remains associated with poor survival outcomes due to its late-stage diagnosis, aggressive tumor biology, and high resistance to conventional therapies [3,4]. The disease is often asymptomatic in its early stages, leading to delayed detection and limited treatment options, contributing to one of the lowest five-year survival rates among all cancers [5]. Given its increasing public health burden, understanding the genomic and clinical factors influencing PC progression is essential for improving risk prediction, early detection, and therapeutic strategies.

While Hispanic/Latino (H/L) individuals have a lower overall incidence of PC compared to Non-Hispanic White (NHW) patients, they are diagnosed at younger ages and at more advanced disease stages, leading to significantly poorer survival outcomes [6]. Additionally, the high prevalence of diabetes mellitus in the H/L population—a known risk factor for PC—has been linked to an increased likelihood of developing the disease [7,8]. However, genomic profiling of PC in H/L patients has been historically limited, creating a critical gap in understanding molecular drivers of disease progression and treatment response. Prior studies have identified actionable somatic mutations in PC that may inform precision medicine approaches, but the extent to which these mutations differ between racial/ethnic groups remains unclear [9,10].

PC is characterized by frequent alterations in key oncogenic pathways that regulate essential cellular functions, such as proliferation, apoptosis, DNA repair, and immune evasion [11,12]. In this study, we focus on five core pathways—TGF-Beta, RTK/RAS, WNT, PI3K, and TP53—selected based on their well-established biological relevance to PC pathogenesis and their high frequency of alteration reported in large-scale datasets, including The Cancer Genome Atlas (TCGA) [13] and other published genomic studies [14,15].

The TGF-Beta signaling pathway has a dual role in PC, acting as a tumor suppressor in early stages but promoting epithelial-to-mesenchymal transition (EMT), invasion, and metastasis in advanced disease [16]. In PC, mutations in SMAD4 and TGFBR2 are commonly observed and have been implicated in disease progression and therapy resistance [17].

Building on this, the RTK/RAS pathway plays a central role in PC pathogenesis, with KRAS mutations occurring in over 90% of cases, driving uncontrolled cell growth and conferring resistance to targeted therapies [18,19]. Mutations in additional RTK/RAS pathway genes, including *ERBB4*, *ALK*, and *HRAS*, may further contribute to tumor progression and therapeutic resistance [20,21].

In parallel, the PI3K/AKT pathway is a critical regulator of tumor metabolism and survival, with frequent alterations in *PTEN* and *PIK3CA* driving increased tumor invasiveness and immune evasion [22]. Mutations in PI3K pathway genes have been associated with poor prognosis and treatment resistance, making them important targets for precision medicine approaches [23,24,25].

Equally critical, the TP53 pathway plays a fundamental role in genomic stability, apoptosis, and cell cycle regulation. *TP53* mutations are among the most common alterations in PC, and emerging evidence suggests that mutant *p53* interacts with *KRAS* to drive metastasis [26,27]. In addition to *TP53* mutations, *CDKN2A* deletions are frequently observed in PC and are associated with aggressive tumor behavior [28,29].

Finally, the WNT/β-catenin signaling pathway is another key driver of PC progression, regulating cell proliferation and differentiation. WNT pathway mutations have been identified in nearly all PC cases, with alterations in genes such as *CTNNB1* and *RNF43* contributing to tumor initiation and progression [30]. Aberrant WNT signaling has also been linked to chemoresistance, underscoring the need for pathway-specific therapeutic interventions [31].

Given the rising incidence of PC and the disproportionately poor outcomes observed in H/L patients, a comprehensive genomic analysis of these key oncogenic pathways is critical for advancing risk prediction, precision medicine, and targeted therapeutic strategies, as has been suggested for other cancers [32,33,34,35,36]. This study aims to characterize pathway-specific mutations in TP53, WNT, PI3K, TGF-Beta, and RTK/RAS signaling among H/L and NHW patients, assess tumor mutation burden, and evaluate the prognostic implications of these alterations. By identifying ethnicity-associated molecular differences, this work seeks to advance clinical epidemiology, risk prediction models, and personalized treatment strategies in underrepresented populations.

## 2. Results

### 2.1. Cohort Composition and Clinical Characteristics of H/L and NHW Pancreatic Cancer Patients

From the cBioPortal projects that reported ethnicity, we identified and constructed our H/L cohort, which included 407 samples, while the NHW cohort comprised 3841 samples (Table 1). The gender distribution was similar between cohorts, with the H/L group consisting of 53.1% male and 46.9% female patients and the NHW group comprising 52.9% male and 47.1% female patients. All patients in both cohorts were diagnosed with primary tumors, with no cases classified as metastatic at the time of data collection. Within the H/L cohort, 95.8% of patients specifically identified as Spanish/Hispanic, while 2.7% were categorized as Spanish NOS, Hispanic NOS, or Latino NOS and 1.5% identified broadly as Hispanic or Latino. In contrast, all patients in the NHW cohort were classified as Non-Hispanic White, ensuring clear ethnic stratification for comparative analysis. As a note, tumor site annotation was available in the dataset, classifying samples as either primary or metastatic. However, detailed information regarding tumor stage and histological grade was not provided. Thus, we were unable to include these established prognostic factors in our analysis. In addition, the cohort primarily consisted of pancreatic adenocarcinoma cases, which accounted for the majority of tumors analyzed. A small number of pancreatic neuroendocrine tumors were also included and are specified in Table 1.

### 2.2. Ethnicity-Associated Differences in Genomic Features of Pancreatic Cancer

The comparative analysis of genomic features between H/L and NHW PC patients reveals significant differences (Table 2). The median mutation count was lower in the H/L cohort compared to the NHW cohort, with a highly significant *p*-value (*p* = 1.02 × 10^−7^), suggesting potential differences in genomic instability between these populations. However, the median tumor mutational burden (TMB) was similar between both groups, indicating that overall mutation rates may not be a key differentiating factor in tumor biology. In contrast, the median fraction of the genome altered (FGA), a measure of chromosomal instability, showed a trend toward being higher in H/L patients, though not statistically significantly so. Stratification by Oncotree classification showed that pancreatic adenocarcinoma (PAAD) was the most prevalent diagnosis in both cohorts, with a similar distribution across histological subtypes. However, notable differences were observed in specific oncogenic mutations. Within the TGF-Beta pathway, *SMAD2* mutations were significantly more prevalent in H/L patients (1.5%, *n* = 6) compared to NHW patients (0.4%, *n* = 14) (*p* = 6.3 × 10^−3^). Similarly, *SMAD4* mutations were detected in 15.0% (*n* = 61) of H/L patients versus 19.9% (*n* = 765) of NHW patients (*p* = 0.02). In the RTK/RAS pathway, mutations in *ERBB4* (3.4%, *n* = 14 vs. 1.8%, *n* = 70; *p* = 0.03), *ALK* (2.7%, *n* = 11 vs. 1.1%, *n* = 42; *p* = 0.01), *HRAS* (1.2%, *n* = 5 vs. 0.1%, *n* = 5; *p* = 1.0 × 10^−4^), and *RIT1* (0.7%, *n* = 3 vs. 0.1%, *n* = 5; *p* = 0.03) were significantly enriched in the H/L cohort. Furthermore, within the WNT pathway, *CTNNB1* mutations were observed in 2.9% (*n* = 12) of H/L patients compared to 1.3% (*n* = 51) of NHW patients (*p* = 0.01). These findings highlight exploratory patterns of ethnicity-associated molecular variation in pancreatic cancer, suggesting that Hispanic/Latino patients may exhibit distinct genomic profiles. While preliminary, these observations could help inform future studies on tumor progression and therapeutic response. Understanding these disparities is crucial for advancing precision medicine strategies tailored to underrepresented populations.

### 2.3. Comparative Analysis of Pathway-Specific Genomic Alterations Reveals Ethnicity-Associated TGF-Beta Differences

In our comparative analysis of pathway-specific genomic alterations in PC among H/L and NHW patients, we identified significant differences in the TGF-Beta pathway, while other pathways exhibited similar alteration frequencies between groups (Table 3). TGF-Beta pathway alterations were significantly less prevalent in H/L patients (18.4%, *n* = 75) compared to NHW patients (24.4%, *n* = 937; *p* = 0.0086), suggesting potential ethnic-specific differences in pathway dysregulation. Conversely, the absence of TGF-Beta pathway alterations was more frequent in the H/L cohort (81.6%, *n* = 332) compared to NHW patients (75.6%, *n* = 2904). Alterations in the RTK/RAS pathway were highly prevalent in both groups, with 85.7% (*n* = 349) of H/L patients and 85.0% (*n* = 3263) of NHW patients exhibiting mutations (*p* = 0.722), indicating a shared molecular signature across both populations. Similarly, no significant difference was observed in WNT pathway alterations, which were present in 11.1% (*n* = 45) of H/L patients and 8.5% (*n* = 325) of NHW patients (*p* = 0.1334), suggesting a potential trend that warrants further investigation. PI3K pathway alterations were detected in 11.3% (*n* = 46) of H/L patients and 11.4% (*n* = 438) of NHW patients (*p* = 1), indicating no substantial disparity between the cohorts. TP53 alterations, which are a hallmark of PC, were detected in 69.0% (*n* = 281) of H/L patients and 70.3% (*n* = 2700) of NHW patients (*p* = 0.6396), showing no significant differences between groups. These findings highlight key molecular distinctions in PC, particularly the lower prevalence of TGF-Beta pathway alterations in H/L patients, which may have implications for disease progression, tumor microenvironment interactions, and therapeutic response. Further research is warranted to elucidate the functional impact of these genomic differences and their relevance to ethnicity-specific clinical outcomes.

### 2.4. Survival Impact of Pathway-Specific Genomic Alterations in H/L Pancreatic Cancer Patients

Kaplan–Meier survival analysis for H/L PC patients with TGF-Beta pathway alterations showed no statistically significant difference in OS between those with and without alterations (*p* = 0.47; Figure 1). The overlapping survival curves and broad confidence intervals suggest that TGF-Beta pathway mutations may not strongly impact prognosis in this cohort. However, the relatively small sample size may limit statistical power, warranting further investigation in larger datasets. Similarly, for the RTK/RAS pathway, no significant difference in OS was observed between patients with and without alterations (*p* = 0.39; Figure 1). While those with mutations had a gradual survival decline, those without maintained higher survival probabilities. Confidence intervals remained wide, indicating variability likely driven by the limited sample size. These results suggest that RTK/RAS alterations may not be reliable prognostic indicators in this group. Kaplan–Meier analysis of WNT pathway alterations also showed no significant association with OS (*p* = 0.89; Figure 1). Patients with WNT alterations exhibited a steady decline in survival, while those without showed slightly improved outcomes. The broad confidence intervals again highlight variability, emphasizing the need for larger studies to determine the clinical relevance of WNT mutations in this population. For the PI3K pathway, survival analysis revealed no significant OS differences between altered and unaltered groups (*p* = 0.37; Figure 1). Although patients with PI3K mutations showed a more gradual decline, wide confidence intervals suggest high variability, limiting interpretability. Further research is needed to clarify the role of PI3K alterations in H/L PC outcomes. Lastly, TP53 pathway alterations were not associated with significant differences in OS (*p* = 0.39; Figure 1). While patients with TP53 mutations showed a steeper early decline, survival curves later converged. Overlapping confidence intervals again suggest that the small sample size may have obscured the true effects.

Overall, these results indicate that mutations in the studied pathways do not significantly impact OS among H/L PC patients. The consistently broad confidence intervals and overlapping curves highlight substantial variability, likely influenced by the limited sample size. These findings indicate the need for larger, well-annotated datasets to better understand the prognostic impact of pathway-specific mutations in underrepresented populations.

### 2.5. Survival Impact of Pathway-Specific Genomic Alterations in NHW Pancreatic Cancer Patients

Kaplan–Meier survival analysis for TGF-Beta pathway alterations in NHW PC patients showed no statistically significant difference in OS between altered and non-altered groups (*p* = 0.29; Appendix A). The closely aligned survival curves and broad confidence intervals suggest the limited prognostic value of TGF-Beta pathway mutations in this cohort, though variability in survival outcomes indicates underlying heterogeneity that warrants further investigation in larger, more diverse populations. For the RTK/RAS pathway, patients with alterations showed a trend toward reduced survival compared to those without, but this was not statistically significant (*p* = 0.091; Appendix A). The survival curve for the altered group declined more gradually, and broad confidence intervals reflect high variability. These findings suggest that RTK/RAS pathway alterations may not have a strong prognostic impact in NHW PC patients, although larger cohorts may help reveal subgroup-specific effects. Similarly, WNT pathway alterations were not associated with significant OS differences (*p* = 0.55; Appendix A). Survival curves for both groups were nearly identical, with overlapping confidence intervals indicating minimal distinction in outcomes. Despite the biological relevance of WNT signaling, these results suggest limited prognostic value in this population, though further research with larger datasets may help clarify potential roles. In contrast, PI3K pathway alterations were not significantly associated with poorer OS (*p* = 0.18; Appendix A). Patients with alterations showed a more pronounced decline in survival, with clear separation between survival curves. Likewise, TP53 pathway alterations were linked to not significantly worse OS (*p* = 0.072; Appendix A). Patients with TP53 mutations showed a marked decline in survival compared to those without, suggesting a critical role in disease progression.

Overall, these results reveal pathway-specific differences in survival among NHW PC patients. While the TGF-Beta, RTK/RAS, and WNT pathways were not significantly associated with survival outcomes, PI3K and TP53 pathway alterations were linked to poorer prognosis. These findings indicate the importance of comprehensive genomic profiling to inform risk stratification and guide targeted therapies, and they highlight the need for larger, well-annotated cohorts to validate and expand upon these insights.

Survival differences across key oncogenic pathways varied by ancestry among pancreatic cancer patients. In the H/L cohort, WNT pathway alterations were associated with a survival probability of 0.00 at 71.7 months, compared to 0.39 in patients without alterations at 24.5 months (95% CI: 0.20–0.74). For TGF-Beta pathway alterations, survival was 0.30 (95% CI: 0.06–1.00) among altered cases versus 0.34 (95% CI: 0.14–0.81) in non-altered cases. TP53 alterations in H/L patients were associated with a 0.29 survival probability at 71.7 months (95% CI: 0.10–0.78), while non-altered patients had a survival of 0.50 at 7.4 months (95% CI: 0.13–1.00). For the PI3K pathway, survival for non-altered H/L patients was 0.30 at 71.7 months (95% CI: 0.11–0.81), while no long-term survival data were available for the altered group. Similarly, RTK/RAS pathway alterations yielded a survival of 0.27 at 71.7 months (95% CI: 0.10–0.73), with no non-altered group data at that timepoint. Among NHW patients, survival differences were also observed. WNT pathway alterations were linked to a 0.11 survival probability at 87.1 months (95% CI: 0.04–0.33), compared to 0.18 in non-altered patients at 80.9 months (95% CI: 0.14–0.23). For TGF-Beta, survival was 0.17 for both altered (69.9 months, 95% CI: 0.11–0.26) and non-altered groups (87.1 months, 95% CI: 0.13–0.23). TP53-altered patients had a survival of 0.16 at 87.1 months (95% CI: 0.12–0.21), whereas non-altered patients had 0.22 survival at 80.9 months (95% CI: 0.14–0.34). PI3K-altered patients had a survival of 0.09 at 62.8 months (95% CI: 0.02–0.49), while non-altered patients had 0.18 survival at 87.1 months (95% CI: 0.14–0.23). RTK/RAS pathway alterations were associated with a survival of 0.27 at 73.7 months (95% CI: 0.14–0.41), with no final timepoint data available for the non-altered group.

### 2.6. Gene-Level Differences in Pathway-Specific Mutations Between H/L and NHW Pancreatic Cancer Patients

The analysis of pathway-specific gene mutations among H/L and NHW PC patients revealed several differences in mutation frequencies (Appendix A). Within the TGF-Beta pathway, *SMAD2* mutations were more frequent in H/L patients (1.5%) compared to NHW patients (0.4%) (*p* = 0.0064), while *SMAD4* mutations were more common in NHW patients (19.9%) versus H/L patients (15.0%) (*p* = 0.0202). Other genes such as *SMAD3, TGFBR1,* and *TGFBR2* showed no significant differences between groups. In the RTK/RAS pathway, *ERBB4* (3.4% in H/L vs. 1.8% in NHW, *p* = 0.0369) and *ALK* (2.7% in H/L vs. 1.1% in NHW, *p* = 0.0109) mutations were more common in H/L patients. Differences in *MET* mutation frequency did not reach statistical significance (*p* = 0.0671). The remaining genes in this pathway showed low mutation rates across both groups. For the WNT pathway, *CTNNB1* mutations were significantly more frequent in H/L patients (2.9%) compared to NHW patients (1.3%) (*p* = 0.01845). *AXIN1* mutations were higher in H/L patients (1.0% vs. 0.3%) but not statistically significantly so (*p* = 0.07244). Other WNT pathway genes, including *APC* and *RNF43*, showed comparable frequencies. PI3K pathway mutations were generally low in both groups, with no statistically significant differences. *PIK3CA, PTEN, PIK3R1, AKT2,* and *MTOR* showed slight variability between groups, but none reached significance. In the TP53 pathway, *TP53* mutations were found in 68.1% of H/L and 64.7% of NHW patients (*p* = 0.2015). *CDKN2A* and *ATM* mutations were slightly more common in NHW patients (*p* = 0.06467 and *p* = 0.06441, respectively), while other genes like *MDM2* and *CHEK2* had low mutation frequencies across both groups.

### 2.7. Ethnicity-Associated Variation in Mutation Types Across Key Oncogenic Pathways

To further explore the nature of pathway-specific genomic alterations, we analyzed the types of mutations present in key genes across the TP53, WNT, PI3K, TGF-Beta, and RTK/RAS pathways among H/L and NHW patients (Appendix A). Missense mutations were the most common alteration type across both populations, particularly in genes such as *HRAS, ERBB4, CTNNB1, MET,* and *RIT1*, where they accounted for over 75% of observed mutations in both cohorts. In contrast, genes like *SMAD4* and *CDKN2A* displayed a more diverse mutation profile, including a substantial proportion of frame shift deletions, nonsense mutations, and splice site alterations. Notably, H/L patients showed a higher proportion of missense mutations in *SMAD2* (66.7%) and *CTNNB1* (91.7%), while NHW patients exhibited a broader distribution of mutation types in *SMAD4* and *ATM*, including translation start site mutations, which were not observed in the H/L cohort. These differences in mutation type and frequency may have implications for protein function, tumor behavior, and response to targeted therapies.

These findings highlight potential ethnicity-specific trends in *SMAD2, SMAD4, CTNNB1, ERBB4,* and *ALK* mutations between H/L and NHW PC patients. However, the lack of statistical significance in most comparisons underscores the need for larger, well-powered studies to validate these preliminary observations. Further research is necessary to explore their potential relevance to precision medicine and targeted therapy in Hispanic/Latino pancreatic cancer patients. Understanding these molecular patterns may ultimately contribute to hypotheses about ethnicity-specific tumor biology and help inform future investigations aimed at improving outcomes in underrepresented populations.

## 3. Discussion

PC remains one of the most lethal malignancies worldwide, with disparities in incidence, disease progression, and survival outcomes across racial and ethnic groups. While NHW patients have historically been the focus of genomic studies, there is a growing recognition of the need to characterize the molecular landscape of PC in H/L populations. Our study sought to address this gap by systematically analyzing alterations in five key oncogenic pathways—TGF-Beta, RTK/RAS, WNT, PI3K, and TP53—among H/L and NHW PC patients. By identifying ethnicity-specific differences in pathway mutations and assessing their prognostic implications, our findings contribute to a more comprehensive understanding of the molecular epidemiology of PC and its potential applications for risk prediction and precision medicine.

### 3.1. Ethnicity-Specific Molecular Differences in PC

Our results reveal distinct molecular profiles in H/L PC patients compared to NHW patients, particularly within the TGF-Beta, RTK/RAS, and WNT pathways. H/L patients had higher *SMAD2* mutation rates, while *SMAD4* mutations were more common in NHW patients, suggesting possible alternative mechanisms of TGF-Beta pathway dysregulation by ethnicity, potentially involving epigenetic or microenvironmental factors. In the RTK/RAS pathway, *ERBB4* and *ALK* mutations were significantly more frequent in H/L patients, along with enrichment of *HRAS* and *RIT1* mutations—indicating alternative RAS activation routes that may hold therapeutic value beyond KRAS-directed therapies. Similarly, WNT pathway analysis showed a higher prevalence of *CTNNB1* mutations in H/L patients, with a trend toward increased *AXIN1* mutations. These alterations may contribute to tumor aggressiveness and resistance, highlighting the potential of WNT-targeted therapies in this population.

Our findings highlight notable population-specific differences in pathway-associated survival among pancreatic cancer patients, with particularly stark contrasts observed in the WNT and TP53 pathways. Among H/L patients, WNT alterations were associated with complete loss of survival by the final timepoint, suggesting a potentially aggressive disease course in this group. Conversely, NHW patients with WNT alterations exhibited slightly longer survival, though still poor, underscoring the prognostic importance of this pathway across ancestries. TP53 alterations were also associated with reduced survival in both populations; however, H/L patients without alterations exhibited unexpectedly higher short-term survival, possibly reflecting differences in tumor biology or healthcare access. The PI3K and RTK/RAS pathways showed consistently poor outcomes when altered, with limited data on non-altered comparators, indicating the need for deeper investigation into these oncogenic drivers. Interestingly, TGF-Beta pathway survival was comparable between altered and non-altered groups in NHW patients but more variable among H/L patients. These findings suggest that the prognostic significance of specific pathways may differ by ancestry, reinforcing the importance of population-inclusive genomic profiling and survival analysis to guide precision oncology approaches.

To further address the prognostic implications of the identified genomic alterations, we performed gene-specific survival analyses for mutations that showed significant differences in frequency between Hispanic/Latino (H/L) and Non-Hispanic White (NHW) pancreatic cancer patients, including *SMAD2*, *SMAD4*, *CTNNB1*, *ERBB4*, and *ALK*. These exploratory analyses revealed no statistically significant associations between individual gene mutations and overall survival, consistent with our broader pathway-level findings. While *SMAD4* mutations are known to influence disease progression in some contexts, their lack of association with survival in our cohort may reflect the multifactorial nature of pancreatic cancer prognosis and the influence of co-occurring molecular or clinical features. Additionally, the relatively low mutation frequencies in certain genes limited our ability to detect modest survival effects. Nonetheless, these results highlight the potential for ethnic-specific genomic alterations to inform future precision oncology efforts, particularly when integrated with functional analyses and larger, well-powered datasets that can more definitively assess their prognostic relevance.

To place our findings in a broader oncologic context, we compared patterns of pathway-specific genomic alterations observed in PC with those reported in recent ethnicity-focused studies of hepatocellular carcinoma (HCC) [37] and gastric cancer (GC) [38]. Across all three cancer types, common oncogenic pathways—including TP53, WNT, PI3K, TGF-Beta, and RTK/RAS—showed consistent yet ethnicity-specific alterations between H/L and NHW patients. Notably, TGF-Beta pathway genes such as *SMAD2*, *SMAD4*, and *TGFBR2* displayed differential mutation frequencies in all three cancers, suggesting a recurring pattern of dysregulation in this pathway among H/L patients. Similarly, WNT pathway alterations—particularly in *CTNNB1* and *APC*—were enriched in H/L patients with PC and GC, reinforcing their potential role in driving tumor biology in this population. While survival differences were not significant among H/L patients in any of the studies, NHW patients consistently exhibited prognostic associations with TP53 and PI3K pathway mutations. These cross-cancer comparisons support the notion that genomic instability may manifest through distinct molecular signatures across ethnic groups, underscoring the need for precision oncology approaches that consider population-specific tumor biology to improve outcomes in underrepresented populations.

Regarding the potential influence of age on genomic instability, we generated and examined age distribution histograms for both the NHW and H/L patient cohorts (Appendix A). The data confirm a notable age disparity: NHW patients exhibit a unimodal distribution with a peak at 70 years, whereas H/L patients display a broader distribution skewed toward younger ages, with a peak around 65 and a meaningful proportion diagnosed before age 50. This age difference is consistent with epidemiological data showing earlier onset of pancreatic cancer among H/L populations. These findings are particularly relevant given that genomic instability often accumulates with age and may influence mutational burden and pathway alterations. Therefore, the observed differences in mutation frequencies—particularly in *SMAD2, SMAD4,* and *CTNNB1*—may partially reflect age-related biological differences. However, the persistence of ethnicity-specific patterns despite this adjustment highlights the importance of incorporating ancestry- and age-informed molecular profiling in future studies to better understand cancer disparities and guide precision oncology efforts in diverse populations.

Our results suggest pathway-specific molecular differences in PC between H/L and NHW patients. In the TGF-Beta pathway, *SMAD2* mutations were more frequent in H/L patients, while *SMAD4* mutations were more common in NHW patients, indicating potential differences in tumor suppressor pathway disruption. Higher mutation rates of *ERBB4* and *ALK* in H/L patients suggest ethnicity-specific alterations in the RTK/RAS pathway, possibly linked to differential tumor biology or therapeutic response. *CTNNB1* mutations were significantly more frequent in H/L patients, highlighting WNT pathway variability, while overall mutational burden in WNT inhibitors remained low. Although PI3K pathway alterations were infrequent, subtle trends in genes like *MTOR* and *AKT2* may warrant further investigation. TP53 mutation rates were similar between groups, but borderline differences in *CDKN2A* and *ATM* suggest variation in DNA repair mechanisms. These results indicate the importance of inclusive genomic studies to uncover clinically relevant differences across populations and inform precision medicine strategies tailored to underrepresented groups.

### 3.2. Clinical Implications for Risk Prediction and Precision Medicine

Our findings suggest that pathway-specific genomic profiling may offer preliminary insights into pancreatic cancer risk and potential therapeutic targets across diverse populations. The observed higher frequency of *SMAD2, ERBB4, ALK,* and *CTNNB1* mutations in Hispanic/Latino patients may warrant further investigation as potential biomarkers for early detection. Similarly, the differing patterns of *SMAD2* mutations in H/L patients and *SMAD4* mutations in Non-Hispanic White patients could reflect underlying biological differences, though additional studies are needed to explore these ethnicity-associated mechanisms and their clinical relevance.

Kaplan–Meier analyses revealed that PI3K and TP53 alterations were significantly associated with poorer survival in NHW patients but not in H/L patients, supporting the potential of PI3K-targeted therapies in the NHW population. The absence of survival associations in H/L patients suggests that tumor biology in this group may be shaped by non-genomic factors, such as immune environment, metabolism, or lifestyle risks. Integrating multi-omics approaches will be essential to fully understand these complex influences and inform precision medicine for underrepresented populations.

The observed differences in the nature of mutations across key oncogenic pathways between H/L and NHW patients may have important biological and clinical implications. The predominance of missense mutations in both groups, particularly in genes such as *HRAS, ERBB4, CTNNB1,* and *MET*, suggests shared mechanisms of oncogenic activation; however, the mutation spectrum in other genes—such as *SMAD4, ATM,* and *CDKN2A*—was notably more diverse in NHW patients. The presence of frame shift, nonsense, and splice site mutations in these genes, especially among NHW cases, may indicate a greater potential for complete loss of function, which could impact tumor suppressor activity and downstream signaling pathways. In contrast, the higher proportion of missense mutations in H/L patients, particularly within *SMAD2* and *CTNNB1*, may result in more nuanced functional alterations that contribute to tumor progression through different biological mechanisms. Additionally, the absence of translation start site mutations in the H/L cohort highlights potential ethnic differences in mutational patterns that warrant further functional validation. These findings indicate the importance of not only identifying which genes are mutated, but also characterizing the type of mutation, as this can influence both prognostic interpretation and therapeutic strategies in a population-specific context.

### 3.3. Limitations and Future Directions

Although this study offers one of the most comprehensive ethnicity-specific genomic analyses of PC to date, it has limitations. The retrospective design and underrepresentation of H/L patients may have introduced selection bias and limited statistical power, particularly for rare mutations. Future validation in larger, prospectively collected, ethnically balanced cohorts is needed. Additionally, functional studies—such as patient-derived organoids and single-cell sequencing—are essential to elucidate the biological impact of *SMAD2, ERBB4, ALK*, and *CTNNB1* mutations and their roles in tumor heterogeneity, drug resistance, and immune evasion.

While H/L individuals have a lower overall incidence of PC compared to NHW patients, they are disproportionately diagnosed at younger ages and often with more advanced disease. This apparent paradox suggests that incidence alone may not fully capture the clinical burden of PC in this population. Contributing factors may include delayed access to care, lower screening rates, or environmental and lifestyle risks such as high rates of diabetes and obesity that could accelerate tumor progression in H/L patients. Additionally, the younger age at diagnosis may reflect an underlying biological susceptibility that is not yet fully understood, possibly related to distinct genomic or epigenomic profiles. Our findings of specific pathway alterations, including enriched mutations in *SMAD2, ERBB4, ALK*, and *CTNNB1* among H/L patients, support the hypothesis that early-onset PC in this population may follow a unique molecular trajectory. These observations support the hypothesis-generating value of investigating age- and ethnicity-associated molecular drivers, which may guide future research into early detection and the development of more tailored treatment approaches.

A limitation of this study is the absence of tumor stage and grade information in the dataset analyzed. While these are important prognosticators in PC, the dataset only included site classification (primary vs. metastatic). This limitation restricted our ability to assess how molecular alterations may correlate with traditional clinicopathological parameters. Nonetheless, this dataset remains one of the few publicly available resources enabling population-specific genomic analyses, particularly among underrepresented groups.

A constraint of this study is the lack of granular histopathological data within the publicly available dataset used for analysis. Specifically, while we stratified cases based on OncoTree classification and identified pancreatic adenocarcinoma as the predominant diagnosis, the database did not include detailed histological subtyping to distinguish between ductal and acinar forms. Although we were able to report specific cancer types such as adenocarcinoma and acinar cell carcinoma, the inability to confirm ductal versus acinar histology for all cases limited our capacity to perform subtype-specific analyses. This indicates the need for more comprehensive clinical annotations in public genomic datasets to enhance the precision and applicability of molecular findings.

Another limitation of this analysis is the imbalance in sample sizes between H/L and NHW patients, with 407 H/L cases compared to 3841 NHW cases. This discrepancy reflects the current underrepresentation of H/L populations in publicly available PC genomic datasets. While the statistical methods employed, such as chi-squared tests, are appropriate and robust for detecting differences even in unbalanced cohorts, the disparity in sample size may still affect the sensitivity and generalizability of some comparisons. This limitation highlights a critical gap in cancer genomics research and reinforces the need for greater inclusion of diverse populations in future genomic studies to ensure that molecular insights and precision medicine strategies are equitably informed across all patient groups.

As this analysis was designed to be exploratory and hypothesis-generating, our primary objective was to identify preliminary signals of ethnicity-associated differences in pathway-specific genomic alterations in pancreatic cancer. We conducted targeted comparisons across well-defined oncogenic pathways (TP53, WNT, PI3K, TGF-Beta, and RTK/RAS) using a curated set of biologically relevant genes. Given the limited scope and predefined nature of these comparisons, we prioritized sensitivity to uncover potential disparities in underrepresented populations—particularly H/L patients—rather than applying stringent multiple testing corrections that could mask emerging patterns. Future studies with larger cohorts and replication datasets will be essential to validate these findings and assess their statistical robustness using more conservative approaches.

Although our Kaplan–Meier survival analyses did not reveal statistically significant differences in overall survival among H/L pancreatic cancer patients with or without pathway-specific genomic alterations, these findings should be interpreted with caution. The limited sample size, particularly for patients with less frequent mutations, may reduce the statistical power of the analysis and obscure true associations. While we did not conduct a formal power analysis, we acknowledge this as a key limitation and highlight the need for future studies with larger, well-annotated cohorts to more definitively determine the prognostic impact of these alterations in underrepresented populations. Expanding sample sizes will be critical for enhancing statistical confidence and enabling subgroup analyses that can inform precision medicine strategies.

An important consideration in this study is the lack of control for clinical and sociodemographic confounders, such as tumor stage, treatment type, comorbidities, and socioeconomic status. These variables were inconsistently reported or unavailable across the analyzed datasets, which may have impacted both genomic profiles and survival outcomes. Thus, our findings should be viewed as exploratory and hypothesis-generating rather than conclusive. The differences observed between H/L and NHW patients may, in part, be influenced by unmeasured clinical or contextual factors. Moving forward, studies that integrate comprehensive clinical and demographic data alongside genomic information will be critical to more precisely define the molecular contributors to PC disparities and to support equitable implementation of precision oncology strategies.

Finally, incorporating ethnicity-specific genomic data into clinical decision-making algorithms will be essential for achieving precision oncology in PC. Our findings suggest that standard risk prediction models based solely on NHW patient data may not fully capture the unique molecular drivers of PC in H/L patients, highlighting the urgent need for inclusive and representative genomic studies.

## 4. Materials and Methods

For this analysis, we leveraged clinical and genomic data from 14 PC datasets accessed via the cBioPortal database. These datasets included studies categorized under PC, as well as data from the GENIE Cohort v17.0 public dataset. To refine our sample pool, we applied strict inclusion criteria, selecting only patients identified as H/L or NHW. Following dataset selection and filtering, four datasets met all criteria, comprising 407 H/L PC patients. For comparison, 3841 NHW PC patients were included using identical criteria (Table 1 and Table 2). This study represents one of the most comprehensive investigations of TP53, WNT, PI3K, TGF-Beta, and RTK/RAS pathway alterations in an underrepresented population, offering key insights into molecular disparities in PC.

Patients were stratified based on ethnicity (H/L vs. NHW) and further categorized according to the presence or absence of TP53, WNT, PI3K, TGF-Beta, and RTK/RAS pathway alterations. This classification allowed a detailed examination of the interactions between genetic alterations and ethnicity. Table 1 presents the number of patients included in the analysis, providing a breakdown of cases by racial/ethnic group. By comparing mutation prevalence between H/L and NHW patients, this study aims to characterize the molecular differences in PC and their implications for precision medicine and targeted therapies.

For statistical analysis, we performed chi-square tests to evaluate the independence of categorical variables and identify potential associations between ethnicity and pathway-specific alterations. This approach enabled us to explore whether certain molecular alterations appeared more frequently in specific racial/ethnic groups, offering preliminary insights into genomic heterogeneity that may help generate hypotheses about differential treatment responses. All statistical analyses were conducted using R software (version 4.3.2).

To assess overall survival (OS), we employed Kaplan–Meier survival analysis, focusing on the impact of TP53, WNT, PI3K, TGF-Beta, and RTK/RAS pathway alterations. Survival curves were generated to visualize OS probabilities over time, comparing patients based on the presence or absence of these molecular disruptions. The log-rank test was used to identify statistically significant differences between survival curves. Additionally, median survival times were calculated, accompanied by 95% confidence intervals, to ensure the robustness of these estimates. This comprehensive methodological approach provides a nuanced understanding of how pathway-specific genomic alterations may influence prognosis in PC, particularly among H/L patients, and offers critical insights for advancing risk prediction models and precision oncology in diverse populations.

To improve clarity and reproducibility, we defined each of the five oncogenic pathways—TGF-Beta, RTK/RAS, WNT, PI3K, and TP53—based on nonsynonymous mutations in key genes associated with each pathway. For example, TGF-Beta pathway alterations included mutations in *SMAD2*, *SMAD3*, *SMAD4*, *TGFBR1/2*, and *ACVR2A/B*; RTK/RAS alterations included mutations *KRAS*, *HRAS*, *ALK*, *ERBB4*, *MET*, and related genes; WNT alterations included mutations in *APC*, *CTNNB1*, and *AXIN1/2*; PI3K alterations included mutations in *PIK3CA*, *PTEN*, *AKT1–3*, and *MTOR*; and TP53 alterations included mutations in *TP53*, *CDKN2A*, *ATM*, *MDM2*, and *CHEK2*. Gene-level frequencies are provided in the Appendix A, and these definitions were applied consistently across all analyses.

Somatic mutation data for genes within the TP53, WNT, PI3K, TGF-Beta, and RTK/RAS pathways were retrieved from cBioPortal, an open-access resource for exploring multidimensional cancer genomics data. The mutation information was extracted from studies that included clinical and demographic annotations and filtered to include only samples with clearly defined ethnicities. These data were used for downstream comparative analyses between H/L and NHW pancreatic cancer patients.

To complement our analysis of mutation frequencies, we examined the types of genomic alterations in key genes within the TP53, WNT, PI3K, TGF-Beta, and RTK/RAS pathways among H/L and NHW PC patients. Mutation types included frame shift, in-frame, missense, nonsense, splice site, and translation start site mutations, as detailed in Appendix A. Missense mutations were the most common across both groups, particularly in *HRAS, MET, CTNNB1, ERBB4,* and *RIT1*. In contrast, genes like *SMAD4*, *ATM*, and *CDKN2A* showed more diverse mutation profiles, including a notable proportion of truncating and splice site variants. These data provide additional insight into potential functional differences in mutation patterns across populations.

## 5. Conclusions

In summary, our study provides key insights into pathway-specific genomic alterations in PC across diverse racial/ethnic groups. *SMAD2, ERBB4, ALK,* and *CTNNB1* mutations were significantly more frequent in H/L patients, while *SMAD4* loss and PI3K pathway alterations were more prognostically relevant in NHW patients. These findings may serve as a basis for future hypothesis-generating studies focused on ethnicity-specific molecular profiling, with the goal of exploring potential avenues for risk stratification, early detection, and targeted treatment approaches. Future efforts should focus on expanding genomic research in underrepresented populations, integrating multi-omics approaches, and developing ethnicity-specific precision medicine strategies to improve PC outcomes for all patients.

## Figures and Tables

**Figure 1 ijms-26-07695-f001:**
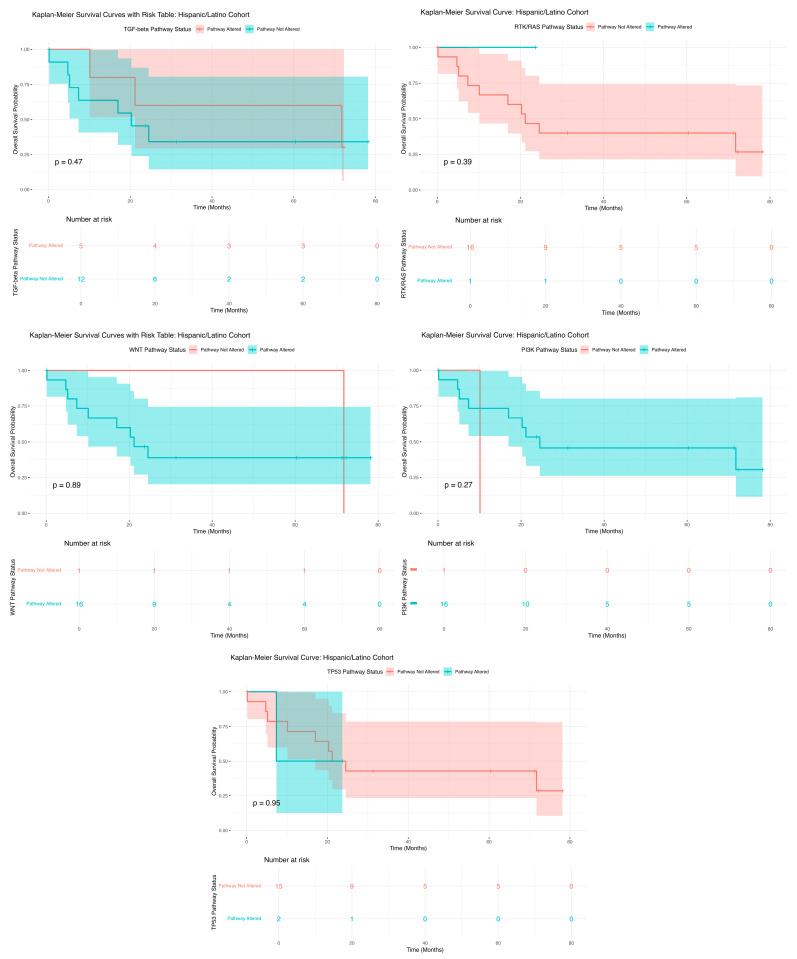
Kaplan–Meier OS curves for Hispanic/Latino (H/L) PC patients, stratified by the presence or absence of TGF-Beta (**upper left**), RTK/RAS (**upper right**), WNT (**middle left**), PI3K (**middle right**), and TP53 (**lower**) pathway alterations.

**Table 1 ijms-26-07695-t001:** Patient demographics and clinical characteristics of the Hispanic/Latino (H/L) and Non-Hispanic White (NHW) PC cohorts.

Clinical Feature	H/L Cohort *n* (%)	NHW Cohort *n* (%)
Gender		
Male	216 (53.1%)	2032 (52.9%)
Female	191 (46.9%)	1809 (47.1%)
Sample Type		
Primary	407 (100.0%)	3841 (100.0%)
Metastatic	0 (0.0%)	0 (0.0%)
Ethnicity		
Spanish/Hispanic	390 (95.8%)	0 (0.0%)
Spanish NOS; Hispanic NOS; Latino NOS	11 (2.7%)	0 (0.0%)
Hispanic or Latino	6 (1.5%)	0 (0.0%)
Non-Spanish; Non-Hispanic	0 (0.0%)	3841 (100.0%)
Cancer Type Detailed		
Acinar Cell Carcinoma of the Pancreas	5 (1.2%)	36 (0.9%)
Adenosquamous Carcinoma of the Pancreas	5 (1.2%)	68 (1.8%)
Cystic Tumor of the Pancreas	0 (0.0%)	1 (0.0%)
Intraductal Papillary Mucinous Neoplasm	1 (0.2%)	27 (0.7%)
Mucinous Cystic Neoplasm	0 (0.0%)	4 (0.1%)
Osteoclastic Giant Cell Tumor	0 (0.0%)	1 (0.0%)
Pancreas	1 (0.2%)	28 (0.7%)
Pancreatic Adenocarcinoma	368 (90.4%)	3390 (88.3%)
Pancreatic Neuroendocrine Tumor	23 (5.7%)	243 (6.3%)
Pancreatoblastoma	0 (0.0%)	4 (0.1%)
Solid Pseudopapillary Neoplasm of the Pancreas	2 (0.5%)	8 (0.2%)
Undifferentiated Carcinoma of the Pancreas	2 (0.5%)	31 (0.8%)

**Table 2 ijms-26-07695-t002:** Ethnicity-associated differences in clinical features between Hispanic/Latino (H/L) and Non-Hispanic White (NHW) pancreatic cancer (PC) cohorts.

Clinical Feature	H/L Samples *n* (%)	NHW Samples *n* (%)	*p*-Value
Median Mutation Count (IQR) *	3 (2–6)	4 (3–7)	1.02 × 10^−7^
Median TMB (IQR) **	2.28 (1.52–3.67)	2.28 (1.52–3.67)	0.8147
Median FGA (IQR) ***	0.043 (0.004–0.176)	0.024 (0.002–0.141)	0.09738
Oncotree Code			
IPMN	1 (0.2%)	27 (0.7%)	0.32
MCN	0 (0.0%)	4 (0.1%)	
OSGCT	0 (0.0%)	1 (0.0%)	
PAAC	5 (1.2%)	36 (0.9%)	
PAAD	368 (90.4%)	3390 (88.3%)	
PAASC	5 (1.2%)	68 (1.8%)	
PACT	0 (0.0%)	1 (0.0%)	
PANCREAS	1 (0.2%)	28 (0.7%)	
PANET	23 (5.7%)	243 (6.3%)	
PB	0 (0.0%)	4 (0.1%)	
SPN	2 (0.5%)	8 (0.2%)	
UCP	2 (0.5%)	31 (0.8%)	
SMAD2 Mutation			
Present	6 (1.5%)	14 (0.4%)	0.006351
Absent	401 (98.5%)	3827 (99.6%)	
SMAD4 Mutation			
Present	61 (15.0%)	765 (19.9%)	0.02016
Absent	346 (85.0%)	3076 (80.1%)	
ERBB4 Mutation			
Present	14 (3.4%)	70 (1.8%)	0.0369
Absent	393 (96.6%)	3771 (98.2%)	
ALK Mutation			
Present	11 (2.7%)	42 (1.1%)	0.01089
Absent	396 (97.3%)	3799 (98.9%)	
HRAS Mutation			
Present	5 (1.2%)	5 (0.1%)	0.000139
Absent	402 (98.8%)	3836 (99.9%)	
RIT1 Mutation			
Present	3 (0.7%)	5 (0.1%)	0.03391
Absent	404 (99.3%)	3836 (99.9%)	
CTNNB1 Mutation			
Present	12 (2.9%)	51 (1.3%)	0.01845
Absent	395 (97.1%)	3790 (98.7%)	

* H/L NA: 16, NHW NA: 168. ** H/L NA: 391, NHW NA: 3385. *** H/L NA: 236, NHW NA: 1483.

**Table 3 ijms-26-07695-t003:** Rates of TGF-Beta, RTK/RAS, WNT, PI3K, and TP53 pathway alterations among Hispanic/Latino (H/L) and Non-Hispanic White (NHW) pancreatic cancer (PC) patients.

	H/L Samples *n* (%)	NHW Samples *n* (%)	*p*-Value
TGF-Beta Alterations Present	75 (18.4%)	937 (24.4%)	0.008641
TGF-Beta Alterations Absent	332 (81.6%)	2904 (75.6%)	
RTK/RAS Alterations Present	349 (85.7%)	3263 (85.0%)	0.722
RTK/RAS Alterations Absent	58 (14.3%)	578 (15.0%)	
WNT Alterations Present	45 (11.1%)	325 (8.5%)	0.1334
WNT Alterations Absent	362 (88.9%)	3416 (88.9%)	
PI3K Alterations Present	46 (11.3%)	438 (11.4%)	1
PI3K Alterations Absent	361 (88.7%)	3403 (88.6%)	
TP53 Alterations Present	281 (69.0%)	2700 (70.3%)	0.6396
TP53 Alterations Absent	126 (31.0%)	1141 (29.7%)	

## Data Availability

All data used in the present study are publicly available at https://www.cbioportal.org/ (accessed on 15 March 2025) and https://genie.cbioportal.org (accessed on 15 March 2025). Additional data can be provided upon reasonable request to the authors.

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
