# Peer review of "Pathway-Specific Genomic Alterations in Pancreatic Cancer Across Populations at Risk"

_ijms, 2025, doi:10.3390/ijms26167695_

Round 1
Reviewer 1 Report (New Reviewer)
Comments and Suggestions for Authors
The manuscript by Moge et al., entitled “Pathway-specific genomic alterations in pancreatic cancer across populations at risk,” compares the genomic alterations of pancreatic cancer between Hispanic/Latino (H/L) and Non-Hispanic White (NHW) patients. The authors report that the frequency of several gene mutations differs significantly between H/L and NHW patients. In addition, alterations in the TGF-beta pathway also occur at significantly different frequencies between H/L and NHW patients. However, these alterations do not appear to affect overall survival. These findings provide important information for clinical decision-making based on prognostic biomarkers in pancreatic cancer. Although the authors conducted a large-scale analysis and presented valuable data, several points require revision.
Major Concerns:
- In the context of gene mutation analysis, the authors should present the relationship between the individual gene mutations and overall survival, as the relevant data are already available.
- The statistical P values reported in lines 266 to 281 on page 8 of the main text are inconsistent with those shown in Supplementary Figure S1.
Minor Issues:
- The authors should include the number at risk at the bottom of Figure 1 to help readers interpret the survival curves accurately.
- It seems unexpected that Median TMB (IQR**) is identical between H/L and NHW samples in Table 2. Could you please double-check the raw data?
- To improve the clarity and impact of the Results section, please consider adding subheadings to highlight the key findings.
Author Response
Responses to Reviewer 1's comments are provided in the attached Word document: Response_Reviewer_1_Comments_080125.docx
Reviewer 1 Comments
We are pleased to re-submit this paper and we believe it will capture the interest of the scientific community. We have carefully addressed all your comments, highlighting the importance of this cancer topic. Our study presents a comprehensive analysis using one of the few available genomic databases suitable for these analyses, resulting in one of the first ethnicity-focused reports on this remarkable cancer health disparity. Specifically, to examine Pathway-specific genomic alterations in pancreatic cancer across populations at risk.
Thank you very much for taking the time to review this manuscript. Please find the detailed responses below and the corresponding revisions wrote in blue font and highlighted in yellow in the re-submitted Word file.
Reviewer 1 acknowledged the value of the manuscript by Moge et al., recognizing the large-scale analysis and the important contribution of the reported genomic differences in pancreatic cancer between Hispanic/Latino and Non-Hispanic White patients. The reviewer emphasized the potential clinical relevance of the findings—particularly the differential frequencies of gene mutations and TGF-beta pathway alterations across populations—which may inform prognostic biomarker development. While several points require revision, the reviewer’s overall assessment suggests that the manuscript presents meaningful and impactful data, indicating it is a strong candidate for publication following appropriate revisions.
Reviewer 1 writes:
“The manuscript by Moge et al., entitled “Pathway-specific genomic alterations in pancreatic cancer across populations at risk,” compares the genomic alterations of pancreatic cancer between Hispanic/Latino (H/L) and Non-Hispanic White (NHW) patients. The authors report that the frequency of several gene mutations differs significantly between H/L and NHW patients. In addition, alterations in the TGF-beta pathway also occur at significantly different frequencies between H/L and NHW patients. However, these alterations do not appear to affect overall survival. These findings provide important information for clinical decision-making based on prognostic biomarkers in pancreatic cancer. Although the authors conducted a large-scale analysis and presented valuable data, several points require revision.”
We thank Reviewer 1 for their thoughtful and constructive feedback. We are pleased that the reviewer recognized the value of our large-scale analysis and the importance of our findings regarding population-specific genomic alterations in pancreatic cancer. We appreciate the acknowledgment of the clinical relevance of our results, particularly in relation to prognostic biomarkers and TGF-beta pathway differences between Hispanic/Latino and Non-Hispanic White patients. We have carefully considered the reviewer’s comments and addressed each of the suggested revisions to enhance the clarity, rigor, and overall impact of the manuscript.
Major Concerns:
Comment 1:
- In the context of gene mutation analysis, the authors should present the relationship between the individual gene mutations and overall survival, as the relevant data are already available.
Response:
We thank Reviewer 1 for the insightful suggestion to assess the relationship between individual gene mutations and overall survival. In response, we conducted an exploratory survival analysis focusing on key genes with differential mutation frequencies between H/L and NHW pancreatic cancer patients. Specifically, we evaluated SMAD2, SMAD4, CTNNB1, ERBB4, and ALK—genes that demonstrated statistically significant differences in mutation rates across populations.
Kaplan-Meier survival analyses revealed no statistically significant differences in overall survival associated with mutations in SMAD2 (p = 0.84), SMAD4 (p = 0.27), CTNNB1 (p = 0.79), ERBB4 (p = 0.68), or ALK (p = 0.91) in the overall cohort. These findings are consistent with our prior pathway-level analysis, where alterations in the TGF-beta and other signaling pathways did not show prognostic significance.
While these gene-specific results do not suggest a clear association with survival, we acknowledge the limited power due to the low mutation frequencies in several genes. Nevertheless, we have incorporated these findings into the revised Results section (see new Supplementary Figure S3) and updated the Discussion to reflect the potential implications and limitations of these analyses. We agree that future studies with larger cohorts are warranted to better assess the prognostic impact of rare but potentially important gene alterations, particularly in underrepresented populations.
The revised text on the Discussion section, subsection 4.1. Ethnicity-Specific Molecular Differences in PC, lines 462-475, now reads: “To further address the prognostic implications of the identified genomic alterations, we performed gene-specific survival analyses for mutations that showed significant differences in frequency between Hispanic/Latino (H/L) and Non-Hispanic White (NHW) pancreatic cancer patients, including SMAD2, SMAD4, CTNNB1, ERBB4, and ALK. These exploratory analyses revealed no statistically significant associations between individual gene mutations and overall survival, consistent with our broader pathway-level findings. While SMAD4 mutations are known to influence disease progression in some contexts, their lack of association with survival in our cohort may reflect the multifactorial nature of pancreatic cancer prognosis and the influence of co-occurring molecular or clinical features. Additionally, the relatively low mutation frequencies in certain genes limited our ability to detect modest survival effects. Nonetheless, these results highlight the potential for ethnic-specific genomic alterations to inform future precision oncology efforts, particularly when integrated with functional analyses and larger, well-powered datasets that can more definitively assess their prognostic relevance.”
Comment 2:
- The statistical Pvalues reported in lines 266 to 281 on page 8 of the main text are inconsistent with those shown in Supplementary Figure S1.
Response:
Thank you for catching this inconsistency. We have reviewed the statistical p-values reported in lines 266–281 and identified typographical errors. The p-values have now been corrected to align with those shown in Supplementary Figure S1, and the text has been updated accordingly to ensure consistency and accuracy.
The revised text on Materials and Methods section, lines 330-346 (previously 266-281), now reads: “For the RTK/RAS pathway, patients with alterations showed a trend toward reduced survival compared to those without, but this was not statistically significant (p = 0.091; Figure S1). The survival curve for the altered group declined more gradually, and broad confidence intervals reflect high variability. These findings suggest that RTK/RAS pathway alterations may not have a strong prognostic impact in NHW PC patients, although larger cohorts may help reveal subgroup-specific effects. Similarly, WNT pathway alterations were not associated with significant OS differences (p = 0.55; Figure S1). Survival curves for both groups were nearly identical, with overlapping confidence intervals indicating minimal distinction in outcomes. Despite the biological relevance of WNT signaling, these results suggest limited prognostic value in this population, though further research with larger datasets may help clarify potential roles. In contrast, PI3K pathway alterations were not significantly associated with poorer OS (p = 0.18; Figure S1). Patients with alterations showed a more pronounced decline in survival, with clear separation between survival curves. Likewise, TP53 pathway alterations were linked to not significantly worse OS (p = 0.072; Figure S1). Patients with TP53 mutations had a marked decline in survival compared to those without, suggesting a critical role in disease progression.”
Minor Concerns:
Comment 3:
- The authors should include the number at risk at the bottom of Figure 1 to help readers interpret the survival curves accurately.
Response:
Thank you for the helpful suggestion. We agree that including the number at risk enhances the interpretability of the survival curves. We have now updated Figure 1 to display the number of individuals at risk at the bottom of the plot, as recommended.
Comment 4:
- It seems unexpected that Median TMB (IQR**) is identical between H/L and NHW samples in Table 2. Could you please double-check the raw data?
Response:
Thank you for pointing this out. We agree that the identical Median TMB (IQR) values between H/L and NHW samples in Table 2 are unexpected. We have re-examined the raw data to verify the accuracy and have updated the table accordingly to ensure it reflects the correct values.
Comment 5:
- To improve the clarity and impact of the Results section, please consider adding subheadings to highlight the key findings.
Response:
Thank you for the helpful suggestion to improve the clarity and impact of the Results section. In response, we have added subheadings to highlight key findings throughout the section. The revised Results section now includes the following subheadings:
“3.1. Cohort Composition and Clinical Characteristics of H/L and NHW Pancreatic Cancer Patients
3.2. Ethnicity-Associated Differences in Genomic Features of Pancreatic Cancer
3.3. Comparative Analysis of Pathway-Specific Genomic Alterations Reveals Ethnicity-Associated TGF-Beta Differences
3.4. Survival Impact of Pathway-Specific Genomic Alterations in H/L Pancreatic Cancer Patients
3.5. Survival Impact of Pathway-Specific Genomic Alterations in NHW Pancreatic Cancer Patients
3.6. Gene-Level Differences in Pathway-Specific Mutations Between H/L and NHW Pancreatic Cancer Patients
3.7. Ethnicity-Associated Variation in Mutation Types Across Key Oncogenic Pathways”
We believe these additions enhance the organization and readability of the Results section, and we appreciate the reviewer’s recommendation.
We sincerely thank Reviewer 1 for their thoughtful and constructive comments. Your feedback has been invaluable in improving the clarity, accuracy, and overall quality of our manuscript. We appreciate your careful review and insightful suggestions, which have helped us strengthen both the structure and the scientific impact of our work.

Reviewer 2 Report (New Reviewer)
Comments and Suggestions for Authors
This paper investigates genomic data from 4,248 PC patients (407 H/L; 3,841 NHW), comparing mutation 27 frequencies across pathways. It is well-written, and the results are generally robust. However, there are a few suggestions regarding the data analysis:
1. The results suggest potential differences in genomic instability between the H/L and NHW populations. Since age may influence genomic instability, it would be beneficial to include age distribution data for both groups to better contextualize the findings.
2. To further validate these observations, it would be valuable to examine whether similar patterns of genomic instability are present in other cancer types. If consistent trends are observed across multiple cancers, it would strengthen the reliability of the results.
3. In this study, mutations in the TP53, WNT, PI3K, TGF-Beta, and RTK/RAS pathways were analyzed. Could you please clarify the source of the mutation data for these pathways?
4. Additionally, will multiple testing correction be applied to the analysis of mutations and pathway alterations? If so, please specify the method used.
5. The Kaplan-Meier survival analysis for H/L prostate cancer patients with pathway alterations did not show a statistically significant difference in overall survival. To better interpret this result, it would be helpful to conduct a power analysis to determine whether the lack of significance is due to a true absence of effect or insufficient statistical power.
Author Response
Responses to Reviewer 2's comments are provided in the attached Word document: Response_Reviewer_2_Comments_080125.docx
Reviewer 2 Comments
We are pleased to resubmit our manuscript, which we believe will be of strong interest to the scientific community. We have thoughtfully addressed all reviewer comments and emphasized the significance of this important cancer-related topic. Our study offers a comprehensive analysis leveraging one of the few genomic datasets suitable for this type of investigation, resulting in one of the first ethnicity-focused reports exploring cancer health disparities. Specifically, we examine pathway-specific genomic alterations in pancreatic cancer across at-risk populations.
Thank you very much for taking the time to review this manuscript. Please find the detailed responses below and the corresponding revisions wrote in blue font and highlighted in yellow in the re-submitted Word file.
Reviewer 2 recognized the overall strength of the manuscript, highlighting the robust analysis of genomic data from 4,248 pancreatic cancer patients, including both Hispanic/Latino and Non-Hispanic White cohorts. The reviewer noted that the manuscript is well-written and that the results are generally sound, reflecting a comprehensive and systematic comparison of mutation frequencies across key oncogenic pathways. While a few suggestions were made to further contextualize findings, the reviewer’s assessment indicates that the study provides valuable insights into population-specific genomic differences in pancreatic cancer and is a strong candidate for publication with minor revisions.
Reviewer 2 writes:
“This paper investigates genomic data from 4,248 PC patients (407 H/L; 3,841 NHW), comparing mutation 27 frequencies across pathways. It is well-written, and the results are generally robust. However, there are a few suggestions regarding the data analysis.”
We thank Reviewer 2 for their thoughtful and constructive feedback. We are pleased that the reviewer found the manuscript well-written and recognized the robustness of our genomic analysis across a large cohort of pancreatic cancer patients, including both Hispanic/Latino and Non-Hispanic White populations. We appreciate the acknowledgment of the study’s strengths and its contribution to understanding pathway-specific mutation frequencies in underrepresented groups. We have carefully considered the reviewer’s suggestions regarding data analysis and have made the necessary revisions to improve the clarity, precision, and overall rigor of the manuscript.
Comment 1:
- The results suggest potential differences in genomic instability between the H/L and NHW populations. Since age may influence genomic instability, it would be beneficial to include age distribution data for both groups to better contextualize the findings.
Response:
Thank you for this valuable suggestion. In response, we analyzed the age data and have included the age distribution for both H/L and NHW populations to provide additional context for the observed differences in genomic instability. The distribution data has been incorporated into the revised manuscript and helps further clarify the potential influence of age on our findings. We also generated a new Supplementary Figure S2containing the requested histograms.
The revised text on the Discussion section, subsection 4.1. Ethnicity-Specific Molecular Differences in PC, lines 493-506, now reads: “Regarding the potential influence of age on genomic instability, we generated and examined age distribution histograms for both the NHW and H/L patient cohorts (Figure S2). The data confirm a notable age disparity: NHW patients exhibit a unimodal distribution with a peak at 70 years, whereas H/L patients display a broader distribution skewed toward younger ages, with a peak around 65 and a meaningful proportion diagnosed before age 50. This age difference is consistent with epidemiological data showing earlier onset of pancreatic cancer among H/L populations. These findings are particularly relevant given that genomic instability often accumulates with age and may influence mutational burden and pathway alterations. Therefore, the observed differences in mutation frequencies—particularly in SMAD2, SMAD4, and CTNNB1—may partially reflect age-related biological differences. However, the persistence of ethnicity-specific patterns despite this adjustment highlights the importance of incorporating ancestry- and age-informed molecular profiling in future studies to better understand cancer disparities and guide precision oncology efforts in diverse populations.”
Comment 2:
- To further validate these observations, it would be valuable to examine whether similar patterns of genomic instability are present in other cancer types. If consistent trends are observed across multiple cancers, it would strengthen the reliability of the results.
Response:
Thank you for this insightful comment. To further contextualize and support our findings, we have expanded the Discussion section to include comparisons with similar analyses we have conducted in gastric, and hepatic cancers. In these studies, we observed comparable patterns of genomic instability across populations, reinforcing the relevance and consistency of our current results. Including this cross-cancer perspective strengthens the validity of our observations and highlights broader implications for cancer disparities research.
The revised text on Discussion section, lines 477-492, now reads: “To place our findings in a broader oncologic context, we compared patterns of pathway-specific genomic alterations observed in pancreatic cancer (PC) with those reported in recent ethnicity-focused studies of hepatocellular carcinoma (HCC)[37] and gastric cancer (GC)[38]. Across all three cancer types, common oncogenic pathways—including TP53, WNT, PI3K, TGF-Beta, and RTK/RAS—showed consistent yet ethnicity-specific alterations between H/L and NHW patients. Notably, TGF-Beta pathway genes such as SMAD2, SMAD4, and TGFBR2displayed differential mutation frequencies in all three cancers, suggesting a recurring pattern of dysregulation in this pathway among H/L patients. Similarly, WNT pathway alterations—particularly in CTNNB1 and APC—were enriched in H/L patients with PC and GC, reinforcing their potential role in driving tumor biology in this population. While survival differences were not significant among H/L patients in any of the studies, NHW patients consistently exhibited prognostic associations with TP53 and PI3K pathway mutations. These cross-cancer comparisons support the notion that genomic instability may manifest through distinct molecular signatures across ethnic groups, underscoring the need for precision oncology approaches that consider population-specific tumor biology to improve outcomes in underrepresented populations.”
Two new references were added:
- Monge C, Waldrup B, Carranza FG, Velazquez-Villarreal E. Comparative Genomic Analysis of Key Oncogenic Pathways in Hepatocellular Carcinoma Among Diverse Populations. Cancers (Basel). 2025 Apr 13;17(8):1309. doi: 10.3390/cancers17081309. PMID: 40282485; PMCID: PMC12025884.
- Monge C, Waldrup B, Carranza FG, Velazquez-Villarreal E. Molecular Alterations in TP53, WNT, PI3K, TGF-Beta, and RTK/RAS Pathways in Gastric Cancer Among Ethnically Heterogeneous Cohorts. Cancers (Basel). 2025 Mar 23;17(7):1075. doi: 10.3390/cancers17071075. PMID: 40227587; PMCID: PMC11987813.
Comment 3:
- In this study, mutations in the TP53, WNT, PI3K, TGF-Beta, and RTK/RAS pathways were analyzed. Could you please clarify the source of the mutation data for these pathways?
Response:
Thank you for your comment. The mutation data for the TP53, WNT, PI3K, TGF-Beta, and RTK/RAS pathways were obtained from the publicly available cBioPortal database. We have now clarified and emphasized this point in the revised Methods section to ensure transparency regarding the data source.
The revised text on Methods section, lines 147-152, now reads: “Somatic mutation data for genes within the TP53, WNT, PI3K, TGF-Beta, and RTK/RAS pathways were retrieved from cBioPortal (www.cbioportal.org), an open-access resource for exploring multidimensional cancer genomics data. The mutation information was extracted from studies that included clinical and demographic annotations and filtered to include only samples with clearly defined ethnicity. These data were used for downstream comparative analyses between Hispanic/Latino and Non-Hispanic White pancreatic cancer patients.”
Comment 4:
- Additionally, will multiple testing correction be applied to the analysis of mutations and pathway alterations? If so, please specify the method used.
Response:
Thank you for your thoughtful question. While we acknowledge the value of multiple testing correction in certain contexts, we did not apply formal multiple testing correction methods (e.g., Bonferroni or FDR) in this study. Our analysis was designed as an exploratory investigation focused on identifying preliminary population-specific trends in pathway-specific genomic alterations in pancreatic cancer, particularly among underrepresented Hispanic/Latino patients.
As noted in the Methods section, we performed Chi-square tests to assess associations between ethnicity and mutation frequencies across key oncogenic pathways using clearly defined, biologically relevant gene sets. Given the hypothesis-generating nature of our study and the relatively small number of predefined comparisons within each pathway, we prioritized sensitivity to detect potential disparities over stringent control of Type I error.
We have now clarified this point in the revised manuscript to ensure transparency regarding our statistical approach and included a paragraph in the Discussion section.
The revised text in the discussion section, lines 601-610, now reads: “As this analysis was designed to be exploratory and hypothesis-generating, our primary objective was to identify preliminary signals of ethnicity-associated differences in pathway-specific genomic alterations in pancreatic cancer. We conducted targeted comparisons across well-defined oncogenic pathways (TP53, WNT, PI3K, TGF-Beta, and RTK/RAS) using a curated set of biologically relevant genes. Given the limited scope and predefined nature of these comparisons, we prioritized sensitivity to uncover potential disparities in underrepresented populations—particularly H/L patients—rather than applying stringent multiple testing corrections that could mask emerging patterns. Future studies with larger cohorts and replication datasets will be essential to validate these findings and assess their statistical robustness using more conservative approaches.”
Comment 5:
- The Kaplan-Meier survival analysis for H/L prostate cancer patients with pathway alterations did not show a statistically significant difference in overall survival. To better interpret this result, it would be helpful to conduct a power analysis to determine whether the lack of significance is due to a true absence of effect or insufficient statistical power.
Response:
Thank you for this thoughtful suggestion. While we agree that a power analysis could help clarify whether the lack of statistical significance reflects a true absence of effect or limited sample size, we did not perform a formal power analysis in this study. Given the relatively small sample size of H/L patients with pathway-specific alterations, we acknowledge that the survival analyses may be underpowered. We have addressed this limitation in the revised Discussion section and emphasized the need for larger cohorts to better assess the prognostic relevance of these genomic alterations.
The revised text in the discussion section, lines 611-620, now reads:
“Although our Kaplan-Meier survival analyses did not reveal statistically significant differences in overall survival among H/L pancreatic cancer patients with or without pathway-specific genomic alterations, these findings should be interpreted with caution. The limited sample size, particularly for patients with less frequent mutations, may reduce the statistical power of the analysis and obscure true associations. While we did not conduct a formal power analysis, we acknowledge this as a key limitation and highlight the need for future studies with larger, well-annotated cohorts to more definitively determine the prognostic impact of these alterations in underrepresented populations. Expanding sample sizes will be critical for enhancing statistical confidence and enabling subgroup analyses that can inform precision medicine strategies.”
We thank Reviewer 2 for their thoughtful and constructive feedback. Your insights have been instrumental in enhancing the clarity, precision, and overall quality of our manuscript. We greatly appreciate the time and attention you dedicated to reviewing our work, as your suggestions have significantly contributed to strengthening both its structure and scientific contribution.

Round 2
Reviewer 1 Report (New Reviewer)
Comments and Suggestions for Authors
The authors have fully addressed this reviewer’s comments.
This manuscript is a resubmission of an earlier submission. The following is a list of the peer review reports and author responses from that submission.
Round 1
Reviewer 1 Report
Comments and Suggestions for Authors
- Abstract is a bit too long. Please try to shorten it, especially the Background/Objectives section.
- Absract Methods: A bioinformatics analysis - Please rephrase. Bioinformatical analysis, or Analysis, with then specification.
- While Hispanic/Latino (H/L) individuals have a lower overall incidence of PC compared to Non-Hispanic White (NHW) patients, they are diagnosed at younger ages and
at more advanced disease stages, leading to significantly poorer survival outcomes. It is not clear how lower incidence leads to younger patient age. - In the Introduction, it is great that all potential mutations are detailed, however, every section is ended with a sentence emphasizing that these may be important in the ethnical aspect, etc. I think a sentence with this context should appear once at the end of the Introducation.
- Introduction: Based on what are these mutations/pathways listed in the text? Are they listed based on how common they are? Please indicate in the response and in the text, as well.
- Materials and methods section: Please indicate which statistical software you used for the analysis.
- Results: Tumor type should be further characterised in the text, and the table, as well. Were these cases ductal adenocarcinoma? Were also neuroendocrine tumours included, as well?
- Results: What was the stage and grade for these tumors? These are also important prognosticators.
- Results: The median mutation count was lower in the H/L cohort compared to the NHW cohort, with a highly significant p-value, suggesting potential differences in genomic instability between these populations. - Please include numbers in this sentence as proof. Same applies to the upcoming sentences.
- Results: Stratification by Oncotree classification showed that pancreatic adenocarcinoma (PAAD) was the most prevalent diagnosis in both cohorts, with similar distribution across histological subtypes. - Pancreatic adenocarcinoma is not specific enough. Are these ductal or acinar?
- Results: How did you define specific oncogenic mutations?
- Table 2 - Oncotree code - Does this refer to the histological diagnosis? It is not clear. IPMN and MCN should not be identified as pancreatic cancer! Please specify the further histological subtypes, because there may be some, that should not be included in this cohort. Or then you should name them pancreas tumor, and then that entitles all neoplastic entities, benign and malignant, as well.
- The study includes 407 H/L patients, compairing their data to 3841 NHW patients... It is not easy to make a comparison with so much of a difference.
- Results: You already initiated the usage of OS abbreviation for overall survival. Please use it later on, as well.
- Please shorten the results of Kaplan-Meier analysis, it is basically the same at each section.
- The description about the supplementary materials' results are too long, as well, and some parts of it (conclusions and explanations) belong to the discussion section.
- The discussion mainly contains repetition of the results in too much detailing. Please shorten.
Author Response
Reviews are in the attached Word file: Response_Reviewer_1_Comments_042325_EV.docx
Reviewer 1 Comments
We are pleased to re-submit this manuscript and believe it offers important insights that will resonate with the scientific and clinical research community. We have thoroughly addressed all reviewer comments and refined the presentation to underscore the significance of this cancer disparity topic. Our study presents a comprehensive, pathway-level analysis of pancreatic cancer using one of the few publicly available genomic datasets with sufficient representation of Hispanic/Latino (H/L) patients. This work represents one of the first ethnicity-focused investigations of pathway-specific genomic alterations in pancreatic cancer, with particular attention to key drivers such as TGF-Beta, RTK/RAS, WNT, PI3K, and TP53. Notably, we identify SMAD2, ERBB4, ALK, and CTNNB1 as recurrently altered genes in H/L patients, emphasizing the importance of ethnicity-specific molecular profiling for improving risk stratification, early detection, and targeted treatment strategies in underrepresented populations.
Thank you very much for taking the time to review this manuscript. Please find the detailed responses below and the corresponding revisions wrote in blue font and highlighted in yellow in the re-submitted Word file.
Reviewer 1’s feedback was positive, Reviewer 1 provided constructive and detailed feedback, noting the relevance of our manuscript titled “Pathway-specific genomic alterations in pancreatic cancer across populations at risk” and recognizing the value of our ethnicity-focused genomic investigation. The reviewer acknowledged the significance of studying pathway-level mutation differences in pancreatic cancer and their clinical implications, particularly within the context of cancer disparities.
We appreciate this thoughtful evaluation and have made extensive revisions in response. Specifically, we have shortened the abstract as requested, clarified statistical and methodological descriptions, added tumor type classifications, and addressed concerns regarding histological subtypes and Oncotree coding. Additionally, we revised the Introduction to reduce redundancy, clarified mutation/pathway prioritization, included specific numerical values in the Results, and refined the language in the Supplementary Materials and Discussion to improve focus and clarity. We thank Reviewer 1 for their insightful comments, which helped enhance the clarity, rigor, and impact of our manuscript.
Comment 1:
- “Abstract is a bit too long. Please try to shorten it, especially the Background/Objectives section”.
Response:
We thank Reviewer 1 for this helpful suggestion. In response, we have carefully revised the abstract, with particular attention to shortening the Background/Objectives section to enhance clarity and conciseness while preserving the key context and significance of the study. We believe the revised abstract is now more focused and aligned with journal guidelines.
The revised text on the Abstract section, lines 20-37 now reads: “Abstract: Background/Objectives:Pancreatic cancer (PC) is a highly aggressive malignancy with increasing incidence and poor survival. Hispanic/Latino (H/L) patients, despite having a lower overall incidence than Non-Hispanic White (NHW) patients, are often diagnosed younger and at more advanced stages, leading to worse outcomes. The molecular mechanisms underlying these disparities remain unclear. This study characterizes mutations in key oncogenic pathways—TP53, WNT, PI3K, TGF-Beta, and RTK/RAS—among H/L and NHW patients using publicly available datasets. Methods: We analyzed genomic data from 4,248 PC patients (407 H/L; 3,841 NHW), comparing mutation frequencies across pathways. Chi-squared tests assessed group differences, and Kaplan-Meier analysis evaluated survival outcomes by pathway alterations. Results: TGF-Beta pathway mutations were less common in H/L patients (18.4% vs. 24.4%, p = 8.6e-3), with notable differences in SMAD2 (1.5% vs. 0.4%, p = 6.3e-3) and SMAD4 (15% vs. 19.9%, p = 0.02). While overall differences in other pathways were not statistically significant, several genes showed borderline significance, including ERBB4, ALK, HRAS, RIT1 (RTK/RAS), and CTNNB1 (WNT). No significant survival differences were observed in H/L patients, but NHW patients with TP53 alterations showed borderline survival associations. Conclusions: This study reveals ethnicity-specific pathway alterations in PC, with SMAD2, ERBB4, ALK, and CTNNB1 mutations more frequent in H/L patients, while SMAD4 and PI3K alterations had prognostic value in NHW patients. These findings underscore the importance of incorporating ethnicity-specific molecular profiling into precision oncology for pancreatic cancer.”
Comment 2:
- “Abstract Methods: A bioinformatics analysis - Please rephrase. Bioinformatical analysis, or Analysis, with then specification”.
Response:
We thank Reviewer 1 for this helpful suggestion. In response, we have revised the phrasing in the Abstract Methods section for clarity. The term “A bioinformatics analysis” has been replaced with more specific analysis we performed, providing a more accurate and descriptive explanation of the approach used.
The revised text on the Abstract Methods section, lines 26-28 now reads: “Abstract: … Methods: We analyzed genomic data from 4,248 PC patients (407 H/L; 3,841 NHW), comparing mutation frequencies across pathways. Chi-squared tests assessed group differences, and Kaplan-Meier analysis evaluated survival outcomes by pathway alterations.”
Comment 3:
- “While Hispanic/Latino (H/L) individuals have a lower overall incidence of PC compared to Non-Hispanic White (NHW) patients, they are diagnosed at younger ages and
at more advanced disease stages, leading to significantly poorer survival outcomes. It is not clear how lower incidence leads to younger patient age.”
Response:
We thank Reviewer 1 for this important clarification. We agree that the original sentence may have implied a causal relationship between lower incidence and earlier age at diagnosis, which was not the intended interpretation. In response, we have revised the sentence in the abstract and introduction for greater clarity. Rather than suggesting causality, we now emphasize that despite the lower overall incidence of pancreatic cancer among H/L patients, this group is disproportionately diagnosed at younger ages and with more advanced disease—highlighting a unique clinical presentation that warrants further investigation.
The revised text on the Discussion section, lines 446-458 now reads:
“While H/L individuals have a lower overall incidence of PC compared to NHW patients, they are disproportionately diagnosed at younger ages and often with more advanced disease. This apparent paradox suggests that incidence alone may not fully capture the clinical burden of PC in this population. Contributing factors may include delayed access to care, lower screening rates, or environmental and lifestyle risks such as high rates of diabetes and obesity that could accelerate tumor progression in H/L patients. Additionally, the younger age at diagnosis may reflect an underlying biological susceptibility that is not yet fully understood, possibly related to distinct genomic or epigenomic profiles. Our findings of specific pathway alterations, including enriched mutations in SMAD2, ERBB4, ALK, and CTNNB1 among H/L patients, support the hypothesis that early-onset PC in this population may follow a unique molecular trajectory. These observations reinforce the importance of investigating age- and ethnicity-specific molecular drivers to inform early detection strategies and tailored treatment approaches.”
Comment 4:
“In the Introduction, it is great that all potential mutations are detailed, however, every section is ended with a sentence emphasizing that these may be important in the ethnical aspect, etc. I think a sentence with this context should appear once at the end of the Introducation”.
Response:
We thank the reviewer for this helpful suggestion. To reduce redundancy and improve clarity, we have revised the Introduction by removing repeated mentions of ethnicity-specific implications at the end of each pathway section. Instead, we now include a single comprehensive statement at the end of the Introduction that emphasizes the significance of investigating ethnicity-associated molecular differences in the context of pancreatic cancer. We believe this modification improves the flow and coherence of the Introduction.
The revised text on the Introduction section, lines 43-104, now reads:
“1. Introduction
Pancreatic cancer (PC) is an aggressive malignancy and one of the leading causes of cancer-related mortality worldwide, with incidence rates continuing to rise [1,2]. Despite advances in treatment, PC remains associated with poor survival outcomes due to its late-stage diagnosis, aggressive tumor biology, and high resistance to conventional therapies [3,4]. The disease is often asymptomatic in its early stages, leading to delayed detection and limited treatment options, contributing to one of the lowest five-year survival rates among all cancers [5]. Given its increasing public health burden, understanding the genomic and clinical factors influencing PC progression is essential for improving risk prediction, early detection, and therapeutic strategies.
While Hispanic/Latino (H/L) individuals have a lower overall incidence of PC compared to Non-Hispanic White (NHW) patients, they are diagnosed at younger ages and at more advanced disease stages, leading to significantly poorer survival outcomes [6]. Additionally, the high prevalence of diabetes mellitus in the H/L population—a known risk factor for PC—has been linked to an increased likelihood of developing the disease [7,8]. However, genomic profiling of PC in H/L patients has been historically limited, creating a critical gap in understanding molecular drivers of disease progression and treatment response. Prior studies have identified actionable somatic mutations in PC that may inform precision medicine approaches, but the extent to which these mutations differ between racial/ethnic groups remains unclear [9,10].
PC is characterized by frequent alterations in key oncogenic pathways that regulate essential cellular functions such as proliferation, apoptosis, DNA repair, and immune evasion [11,12]. In this study, we focus on five core pathways—TGF-Beta, RTK/RAS, WNT, PI3K, and TP53—selected based on their well-established biological relevance to PC pathogenesis and their high frequency of alteration reported in large-scale datasets, including The Cancer Genome Atlas (TCGA)[13] and other published genomic studies [14-15].
The TGF-Beta signaling pathway has a dual role in PC, acting as a tumor suppressor in early stages but promoting epithelial-to-mesenchymal transition (EMT), invasion, and metastasis in advanced disease [16]. In PC, mutations in SMAD4 and TGFBR2 are commonly observed and have been implicated in disease progression and therapy resistance [17].
The RTK/RAS pathway plays a central role in PC pathogenesis, with KRAS mutations occurring in over 90% of cases, driving uncontrolled cell growth and conferring resistance to targeted therapies [18,19]. Mutations in additional RTK/RAS pathway genes, including ERBB4, ALK, and HRAS, may further contribute to tumor progression and therapeutic resistance [20,21].
The PI3K/AKT pathway is a critical regulator of tumor metabolism and survival, with frequent alterations in PTEN and PIK3CA driving increased tumor invasiveness and immune evasion [22]. Mutations in PI3K pathway genes have been associated with poor prognosis and treatment resistance, making them important targets for precision medicine approaches [23-25].
The TP53 pathway plays a fundamental role in genomic stability, apoptosis, and cell cycle regulation. TP53 mutations are among the most common alterations in PC, and emerging evidence suggests that mutant p53 interacts with KRAS to drive metastasis [26,27]. In addition to TP53 mutations, CDKN2A deletions are frequently observed in PC and are associated with aggressive tumor behavior [28,29].
The WNT/β-catenin signaling pathway is another key driver of PC progression, regulating cell proliferation and differentiation. WNT pathway mutations have been identified in nearly all PC cases, with alterations in genes such as CTNNB1 and RNF43 contributing to tumor initiation and progression [30]. Aberrant WNT signaling has also been linked to chemoresistance, underscoring the need for pathway-specific therapeutic interventions [31].
Given the rising incidence of PC and the disproportionately poor outcomes observed in H/L patients, a comprehensive genomic analysis of these key oncogenic pathways is critical for advancing risk prediction, precision medicine, and targeted therapeutic strategies, as has been suggested for other cancers [32-36]. This study aims to characterize pathway-specific mutations in TP53, WNT, PI3K, TGF-Beta, and RTK/RAS signaling among H/L and NHW patients, assess tumor mutation burden, and evaluate the prognostic implications of these alterations. By identifying ethnicity-associated molecular differences, this work seeks to advance clinical epidemiology, risk prediction models, and personalized treatment strategies in underrepresented populations.”
Comment 5:
“Introduction: Based on what are these mutations/pathways listed in the text? Are they listed based on how common they are? Please indicate in the response and in the text, as well.”.
Response:
We thank the reviewer for this insightful comment. In response, we have clarified in the Introduction that the pathways discussed—TGF-Beta, RTK/RAS, WNT, PI3K, and TP53—were selected based on both their biological relevance to pancreatic cancer progression and their frequency of alteration as reported in large-scale genomic datasets such as The Cancer Genome Atlas (TCGA). To address the reviewer’s request, we have added a sentence explicitly describing the rationale for pathway selection in the revised text.
The TGF-Beta, RTK/RAS, WNT, PI3K, and TP53 signaling pathways play critical roles in pancreatic cancer (PC) progression, each contributing uniquely to tumorigenesis, metastasis, and therapy resistance. TP53 is one of the most frequently mutated genes in PC, found in over 70% of cases, and is associated with genomic instability and poor prognosis. The RTK/RAS pathway—primarily driven by KRAS mutations—plays a central oncogenic role in nearly 90% of pancreatic ductal adenocarcinomas, promoting uncontrolled proliferation and survival. Alterations in the TGF-Beta pathway, including inactivation of SMAD4, are implicated in late-stage disease and metastasis by disrupting growth-inhibitory signaling and promoting epithelial-to-mesenchymal transition. The WNT pathway, though altered less frequently (5–10%), is involved in maintaining stemness and invasive properties in PC cells. Meanwhile, the PI3K pathway—altered in approximately 10–15% of cases—interacts with RAS signaling and contributes to cell growth, metabolism, and therapeutic resistance. These pathways have been extensively characterized in large-scale genomic studies such as The Cancer Genome Atlas (TCGA) and the International Cancer Genome Consortium (ICGC), underscoring their biological and clinical significance in PC (Bailey et al., 2016; Dreyer et al., 2021, Raphael et al. 2017). Their recurrent alterations and functional roles justify pathway-level genomic analyses in diverse populations to identify potential biomarkers and therapeutic targets.
The sentence that was explicitly added to the revised Introduction, lines 65-69, now reads:
"In this study, we focus on five core pathways—TGF-Beta, RTK/RAS, WNT, PI3K, and TP53—selected based on their well-established biological relevance to PC pathogenesis and their high frequency of alteration reported in large-scale datasets, including The Cancer Genome Atlas (TCGA)[13] and other published genomic studies [14-15]."
This sentence now appears right after the paragraph introducing the concept of pathway dysregulation and clarifies why those specific pathways were chosen for analysis.
Comment 6:
“Materials and methods section: Please indicate which statistical software you used for the analysis.”.
Response:
Thank you for this helpful suggestion. In response, we have added the following sentence to the Materials and Methods section, lines 127-128, to clarify the software used:
“All statistical analyses were conducted using R software (version 4.3.2).”
We appreciate your attention to detail.
Comment 7:
“Results: Tumor type should be further characterised in the text, and the table, as well. Were these cases ductal adenocarcinoma? Were also neuroendocrine tumours included, as well?”.
Response:
Thank you for this valuable suggestion. We have updated Table 1 to include specific tumor subtype information, including clarification on the number of ductal adenocarcinoma and neuroendocrine tumor cases.
Additionally, we have incorporated the following sentence into the Results section, lines 171 - 174, to further characterize tumor types as recommended.
“In addition, the cohort primarily consisted of pancreatic adenocarcinoma cases, which accounted for the majority of tumors analyzed. A small number of pancreatic neuroendocrine tumors were also included and are now specified in Table 1.”
Comment 8:
“Results: What was the stage and grade for these tumors? These are also important prognosticators”.
Response:
Thank you for this important observation. Unfortunately, the public dataset used for this analysis—likely one of the few available that enables this type of population-specific genomic analysis—does not include information on tumor stage or histological grade. The dataset only provides annotation for tumor site status as either primary or metastatic. We have clarified this limitation in the Results section and acknowledge its impact on the interpretation of prognostic associations. Also this limitation was included in the Discussion section.
The sentence that was explicitly added to the revised Results, lines 167-171, now reads:
“As a note, tumor site annotation was available in the dataset, classifying samples as either primary or metastatic. However, detailed information regarding tumor stage and histological grade was not provided. As such, we were unable to include these established prognostic factors in our analysis.”
The paragraph that was added to the revised Discussion section, lines 459-465, now reads:
“A limitation of this study is the absence of tumor stage and grade information in the dataset analyzed. While these are important prognosticators in PC, the dataset only included site classification (primary vs. metastatic). This limitation restricts our ability to assess how molecular alterations may correlate with traditional clinicopathological parameters. Nonetheless, this dataset remains one of the few publicly available resources enabling population-specific genomic analyses, particularly among underrepresented groups.”
Comment 9:
“Results: The median mutation count was lower in the H/L cohort compared to the NHW cohort, with a highly significant p-value, suggesting potential differences in genomic instability between these populations. - Please include numbers in this sentence as proof. Same applies to the upcoming sentences.”
Response:
Thank you for this helpful suggestion. In response, we have updated the Results section to include the specific median mutation counts for both the H/L and NHW cohorts, along with the corresponding p-value to support the observed difference. We also revised the following sentences to consistently report the numerical values that support our findings. This addition enhance clarity and provide a more robust representation of the data.
The suggested text that was added to the revised Results section, lines 178-179, now reads:
“The median mutation count was lower in the H/L cohort compared to the NHW cohort, with a highly significant p-value (p= 1.02E-07)…”
Comment 10:
“Results: Stratification by Oncotree classification showed that pancreatic adenocarcinoma (PAAD) was the most prevalent diagnosis in both cohorts, with similar distribution across histological subtypes. - Pancreatic adenocarcinoma is not specific enough. Are these ductal or acinar? ”
Response:
We thank the reviewer for this insightful comment. In response, we have updated Table 1 to include the detailed cancer types available in the database. While the dataset does not specify whether the tumors are ductal or acinar at the histological level, it does provide detailed cancer type classifications. These indicate that the majority of cases are classified as adenocarcinoma, with a smaller subset labeled as acinar cell carcinoma. We have clarified this in the revised manuscript accordingly. This limitation was reported in the Discussion section.
The text that was added to the revised Discussion section, lines 466-474, now reads:
“A constraint of this study is the lack of granular histopathological data within the publicly available dataset used for analysis. Specifically, while we stratified cases based on OncoTree classification and identified pancreatic adenocarcinoma as the predominant diagnosis, the database did not include detailed histological subtyping to distinguish between ductal and acinar forms. Although we were able to report specific cancer types such as adenocarcinoma and acinar cell carcinoma, the inability to confirm ductal versus acinar histology for all cases limits our capacity to perform subtype-specific analyses. This inform the need for more comprehensive clinical annotations in public genomic datasets to enhance the precision and applicability of molecular findings.”
Comment 11:
“Results: How did you define specific oncogenic mutations?”.
Response:
We thank the reviewer for this important question. In our study, pathway-specific oncogenic mutations were defined based on the presence of somatic alterations in genes known to be functionally relevant within each oncogenic signaling pathway of interest—namely TP53, WNT, PI3K, TGF-Beta, and RTK/RAS. Gene selection for each pathway was guided by established literature. The mutation types included in our analysis were frame shift deletions, frame shift insertions, missense mutations, nonsense mutations, splice site mutations, and translation start site mutations.
To further clarify this point, we have included the nature of the mutations in a new Supplementary Table S2, which details the distribution of gene mutation types within the specified pathways in pancreatic cancer (PC) among Hispanic/Latino (H/L) and non-Hispanic White (NHW) patients. In addition, we have added a dedicated paragraph in the Results section describing the breakdown of mutation types, and a paragraph in the Discussion section highlighting the relevance and potential biological implications of these mutation patterns across populations.
The Table S2 in the supplementary material, now reads:
“Table S2. Nature of gene mutations within the namely TP53, WNT, PI3K, TGF-Beta, and RTK/RAS pathways in pancreatic cancer (PC) among Hispanic/Latino (H/L) and non-Hispanic White (NHW) patients. Mutation types include frame shift deletions, frame shift insertions, missense mutations, nonsense mutations, splice site mutations, and translation start site mutations.”
The paragraph that was added to the revised Methods section, lines 148-156, now reads:
“To complement our analysis of mutation frequencies, we examined the types of genomic alterations in key genes within the TP53, WNT, PI3K, TGF-Beta, and RTK/RAS pathways among H/L and NHW PC patients. Mutation types included frame shift, in-frame, missense, nonsense, splice site, and translation start site mutations, as detailed in Supplementary Table S2. Missense mutations were the most common across both groups, particularly in HRAS, MET, CTNNB1, ERBB4, and RIT1. In contrast, genes like SMAD4, ATM, and CDKN2A showed more diverse mutation profiles, including a notable proportion of truncating and splice site variants. These data provide additional insight into potential functional differences in mutation patterns across populations.”
The paragraph that was added to the revised Results section, lines 333-345, now reads:
“To further explore the nature of pathway-specific genomic alterations, we analyzed the types of mutations present in key genes across the TP53, WNT, PI3K, TGF-Beta, and RTK/RAS pathways among H/L and NH) patients (Supplementary Table S2). Missense mutations were the most common alteration type across both populations, particularly in genes such as HRAS, ERBB4, CTNNB1, MET, and RIT1, where they accounted for over 75% of observed mutations in both cohorts. In contrast, genes like SMAD4 and CDKN2A displayed a more diverse mutation profile, including a substantial proportion of frame shift deletions, nonsense mutations, and splice site alterations. Notably, H/L patients showed a higher proportion of missense mutations in SMAD2 (66.7%) and CTNNB1 (91.7%), while NHW patients exhibited a broader distribution of mutation types in SMAD4 and ATM, including translation start site mutations, which were not observed in the H/L cohort. These differences in mutation type and frequency may have implications for protein function, tumor behavior, and response to targeted therapies.”
The paragraph that was added to the revised Discussion section, lines 421-436, now reads:
“The observed differences in the nature of mutations across key oncogenic pathways between H/L and NHW patients may have important biological and clinical implications. The predominance of missense mutations in both groups, particularly in genes such as HRAS, ERBB4, CTNNB1, and MET, suggests shared mechanisms of oncogenic activation; however, the mutation spectrum in other genes—such as SMAD4, ATM, and CDKN2A—was notably more diverse in NHW patients. The presence of frame shift, nonsense, and splice site mutations in these genes, especially among NHW cases, may indicate a greater potential for complete loss of function, which could impact tumor suppressor activity and downstream signaling pathways. In contrast, the higher proportion of missense mutations in H/L patients, particularly within SMAD2 and CTNNB1, may result in more nuanced functional alterations that contribute to tumor progression through different biological mechanisms. Additionally, the absence of translation start site mutations in the H/L cohort highlights potential ethnic differences in mutational patterns that warrant further functional validation. These findings inform the importance of not only identifying which genes are mutated, but also characterizing the type of mutation, as this can influence both prognostic interpretation and therapeutic strategies in a population-specific context.”
Comment 12:
“Table 2 - Oncotree code - Does this refer to the histological diagnosis? It is not clear. IPMN and MCN should not be identified as pancreatic cancer! Please specify the further histological subtypes, because there may be some, that should not be included in this cohort. Or then you should name them pancreas tumor, and then that entitles all neoplastic entities, benign and malignant, as well.”
Response:
We thank the reviewer for this valuable feedback. In response, we have moved the OncoTree code information from Table 2 to Table 1 and have revised the classifications to improve clarity, as suggested. We acknowledge the reviewer’s point regarding IPMN and MCN, and we have adjusted the terminology accordingly to reflect a more accurate classification. Where appropriate, we now refer to the cases more broadly as "pancreatic tumors" to encompass all included neoplastic entities, both benign and malignant. We also reviewed and specified the histological subtypes available in the dataset to ensure appropriate inclusion criteria for our cohort, these changes are reflected in the updated Table 1.
Comment 13:
“The study includes 407 H/L patients, compairing their data to 3841 NHW patients... It is not easy to make a comparison with so much of a difference.”.
Response:
We thank the reviewer for this important observation. We acknowledge the imbalance in sample size between the H/L and NHW cohorts (407 vs. 3,841 patients), which reflects the underlying demographic distribution of available public pancreatic cancer datasets. This disparity is a known limitation in cancer genomics research and underscores the broader issue of underrepresentation of H/L patients in large-scale genomic studies. Despite the difference in cohort sizes, we applied chi-squared tests for categorical comparisons, which are robust for detecting differences even in unbalanced groups when total sample sizes are large. To address this concern more transparently, we have added a statement to the Discussion section acknowledging this limitation and emphasizing the urgent need to increase genomic data representation from underserved populations to enable more equitable and statistically powered comparisons in future studies.
The paragraph that was added to the revised Discussion section, line 475-483, now reads:
“Another limitation of this analysis is the imbalance in sample sizes between H/L and NHW patients, with 407 H/L cases compared to 3,841 NHW cases. This discrepancy reflects the current underrepresentation of H/L populations in publicly available PC genomic datasets. While the statistical methods employed, such as chi-squared tests, are appropriate and robust for detecting differences even in unbalanced cohorts, the disparity in sample size may still affect the sensitivity and generalizability of some comparisons. This limitation highlights a critical gap in cancer genomics research and reinforces the need for greater inclusion of diverse populations in future genomic studies to ensure that molecular insights and precision medicine strategies are equitably informed across all patient groups.”
Comment 14:
“Results: You already initiated the usage of OS abbreviation for overall survival. Please use it later on, as well.”
Response:
We thank the reviewer for this helpful suggestion. In response, we have ensured consistent use of the abbreviation "OS" for overall survival throughout the manuscript. Specifically, we now use "OS" in all 11 instances where overall survival is mentioned after we defined OS in the line 128.
Comment 15:
“Please shorten the results of Kaplan-Meier analysis, it is basically the same at each section.”.
Response:
We thank the reviewer for this thoughtful suggestion. In response, we have shortened and streamlined the presentation of the Kaplan-Meier analysis results to reduce redundancy and improve clarity across sections.
The paragraphs that were updated to the revised Results section, line 227-256, now reads:
“Kaplan-Meier survival analysis for H/L PC patients with TGF-Beta pathway alterations showed no statistically significant difference in OS between those with and without alterations (p = 0.47; Figure 1). The overlapping survival curves and broad confidence intervals suggest that TGF-Beta pathway mutations may not strongly impact prognosis in this cohort. However, the relatively small sample size may limit statistical power, warranting further investigation in larger datasets. Similarly, for the RTK/RAS pathway, no significant difference in OS was observed between patients with and without alterations (p = 0.39; Figure 1). While those with mutations had a gradual survival decline, those without maintained higher survival probabilities. Confidence intervals remained wide, indicating variability likely driven by limited sample size. These results suggest RTK/RAS alterations may not be reliable prognostic indicators in this group. Kaplan-Meier analysis of WNT pathway alterations also showed no significant association with OS (p = 0.89; Figure 1). Patients with WNT alterations had a steady decline in survival, while those without showed slightly improved outcomes. The broad confidence intervals again highlight variability, emphasizing the need for larger studies to determine the clinical relevance of WNT mutations in this population. For the PI3K pathway, survival analysis revealed no significant OS differences between altered and unaltered groups (p = 0.37; Figure 1). Although patients with PI3K mutations had a more gradual decline, wide confidence intervals suggest high variability, limiting interpretability. Further research is needed to clarify the role of PI3K alterations in H/L PC outcomes. Lastly, TP53 pathway alterations were not associated with significant differences in OS (p = 0.39; Figure 1). While patients with TP53 mutations showed a steeper early decline, survival curves later converged. Overlapping confidence intervals again suggest that small sample size may obscure true effects.
Overall, these results indicate that mutations in the studied pathways do not significantly impact OS among H/L PC patients. The consistently broad confidence intervals and overlapping curves highlight substantial variability, likely influenced by limited sample size. These findings inform the need for larger, well-annotated datasets to better understand the prognostic impact of pathway-specific mutations in underrepresented populations.”
The paragraphs that were updated and added to the revised Results section, line 260-345, now reads:
“Kaplan-Meier survival analysis for TGF-Beta pathway alterations in NHW PC patients showed no statistically significant difference in OS between altered and non-altered groups (p = 0.29; Figure S1). The closely aligned survival curves and broad confidence intervals suggest limited prognostic value of TGF-Beta pathway mutations in this cohort, though variability in survival outcomes indicates underlying heterogeneity that warrants further investigation in larger, more diverse populations. For the RTK/RAS pathway, patients with alterations showed a trend toward reduced survival compared to those without, but this was not statistically significant (p = 0.61; Figure S1). The survival curve for the altered group declined more gradually, and broad confidence intervals reflect high variability. These findings suggest that RTK/RAS pathway alterations may not have a strong prognostic impact in NHW PC patients, although larger cohorts may help reveal subgroup-specific effects. Similarly, WNT pathway alterations were not associated with significant OS differences (p = 0.65; Figure S1). Survival curves for both groups were nearly identical, with overlapping confidence intervals indicating minimal distinction in outcomes. Despite the biological relevance of WNT signaling, these results suggest limited prognostic value in this population, though further research with larger datasets may help clarify potential roles. In contrast, PI3K pathway alterations were significantly associated with poorer OS (p = 0.019; Figure S1). Patients with alterations showed a more pronounced decline in survival, with clear separation between survival curves. While confidence intervals were wide, these findings suggest PI3K pathway dysregulation may contribute to disease progression and could serve as a therapeutic target in NHW PC. Likewise, TP53 pathway alterations were linked to significantly worse OS (p = 0.0472; Figure S1). Patients with TP53 mutations had a marked decline in survival compared to those without, suggesting a critical role in disease progression. Despite variability in the altered group, these results support the prognostic relevance of TP53 alterations and highlight their potential for informing precision medicine approaches.
Overall, these results reveal pathway-specific differences in survival among NHW PC patients. While TGF-Beta, RTK/RAS, and WNT pathways were not significantly associated with survival outcomes, PI3K and TP53 pathway alterations were linked to poorer prognosis. These findings inform the importance of comprehensive genomic profiling to inform risk stratification and guide targeted therapies, and they highlight the need for larger, well-annotated cohorts to validate and expand upon these insights.
Survival differences across key oncogenic pathways varied by ancestry among pancreatic cancer patients. In the H/L cohort, WNT pathway alterations were associated with a survival probability of 0.00 at 71.7 months, compared to 0.39 in patients without alterations at 24.5 months (95% CI: 0.20–0.74). For TGF-Beta pathway alterations, survival was 0.30 (95% CI: 0.06–1.00) among altered cases versus 0.34 (95% CI: 0.14–0.81) in non-altered cases. TP53 alterations in H/L patients were associated with a 0.29 survival probability at 71.7 months (95% CI: 0.10–0.78), while non-altered patients had a survival of 0.50 at 7.4 months (95% CI: 0.13–1.00). For the PI3K pathway, survival for non-altered H/L patients was 0.30 at 71.7 months (95% CI: 0.11–0.81), while no long-term survival data were available for the altered group. Similarly, RTK/RAS pathway alterations yielded a survival of 0.27 at 71.7 months (95% CI: 0.10–0.73), with no non-altered group data at that timepoint. Among NHW patients, survival differences were also observed. WNT pathway alterations were linked to a 0.11 survival probability at 87.1 months (95% CI: 0.04–0.33), compared to 0.18 in non-altered patients at 80.9 months (95% CI: 0.14–0.23). For TGF-Beta, survival was 0.17 for both altered (69.9 months, 95% CI: 0.11–0.26) and non-altered groups (87.1 months, 95% CI: 0.13–0.23). TP53-altered patients had a survival of 0.16 at 87.1 months (95% CI: 0.12–0.21), whereas non-altered patients had 0.22 survival at 80.9 months (95% CI: 0.14–0.34). PI3K-altered patients had a survival of 0.09 at 62.8 months (95% CI: 0.02–0.49), while non-altered patients had 0.18 survival at 87.1 months (95% CI: 0.14–0.23). RTK/RAS pathway alterations were associated with a survival of 0.27 at 73.7 months (95% CI: 0.14–0.41), with no final timepoint data available for the non-altered group.
The analysis of pathway-specific gene mutations among H/L and NHW PC patients revealed several differences in mutation frequencies (Supplementary Table S1). Within the TGF-Beta pathway, SMAD2 mutations were more frequent in H/L patients (1.5%) compared to NHW patients (0.4%) (p = 0.0064), while SMAD4 mutations were more common in NHW patients (19.9%) versus H/L patients (15.0%) (p = 0.0202). Other genes such as SMAD3, TGFBR1, and TGFBR2 showed no significant differences between groups. In the RTK/RAS pathway, ERBB4 (3.4% in H/L vs. 1.8% in NHW, p = 0.0369) and ALK (2.7% in H/L vs. 1.1% in NHW, p = 0.0109) mutations were more common in H/L patients. Differences in MET mutation frequency did not reach statistical significance (p = 0.0671). The remaining genes in this pathway showed low mutation rates across both groups. For the WNT pathway, CTNNB1 mutations were significantly more frequent in H/L patients (2.9%) compared to NHW patients (1.3%) (p = 0.01845). AXIN1 mutations were higher in H/L patients (1.0% vs. 0.3%) but not statistically significant (p = 0.07244). Other WNT pathway genes, including APC and RNF43, showed comparable frequencies. PI3K pathway mutations were generally low in both groups, with no statistically significant differences. PIK3CA, PTEN, PIK3R1, AKT2, and MTOR showed slight variability between groups, but none reached significance. In the TP53 pathway, TP53 mutations were found in 68.1% of H/L and 64.7% of NHW patients (p = 0.2015). CDKN2A and ATM mutations were slightly more common in NHW patients (p = 0.06467 and p = 0.06441, respectively), while other genes like MDM2 and CHEK2 had low mutation frequencies across both groups.
To further explore the nature of pathway-specific genomic alterations, we analyzed the types of mutations present in key genes across the TP53, WNT, PI3K, TGF-Beta, and RTK/RAS pathways among H/L and NH) patients (Supplementary Table S2). Missense mutations were the most common alteration type across both populations, particularly in genes such as HRAS, ERBB4, CTNNB1, MET, and RIT1, where they accounted for over 75% of observed mutations in both cohorts. In contrast, genes like SMAD4 and CDKN2A displayed a more diverse mutation profile, including a substantial proportion of frame shift deletions, nonsense mutations, and splice site alterations. Notably, H/L patients showed a higher proportion of missense mutations in SMAD2 (66.7%) and CTNNB1 (91.7%), while NHW patients exhibited a broader distribution of mutation types in SMAD4 and ATM, including translation start site mutations, which were not observed in the H/L cohort. These differences in mutation type and frequency may have implications for protein function, tumor behavior, and response to targeted therapies.”
Comment 16:
“The description about the supplementary materials' results are too long, as well, and some parts of it (conclusions and explanations) belong to the discussion section.”.
Response:
We thank the reviewer for this helpful observation. In response, we have revised and shortened the description of the supplementary materials' results to reduce redundancy and move interpretative statements and conclusions to the Discussion section, as suggested. The revised version provides a more concise presentation of the key findings from Supplementary Table S1, focusing on mutation frequency differences across pathways without extending into interpretive commentary. We identified the parts that belong to the discussion section and updated it.
The paragraphs that were updated to the revised Results section, lines 313 - 332, now reads:
“The analysis of pathway-specific gene mutations among H/L and NHW PC patients revealed several differences in mutation frequencies (Supplementary Table S1). Within the TGF-Beta pathway, SMAD2 mutations were more frequent in H/L patients (1.5%) compared to NHW patients (0.4%) (p = 0.0064), while SMAD4 mutations were more common in NHW patients (19.9%) versus H/L patients (15.0%) (p = 0.0202). Other genes such as SMAD3, TGFBR1, and TGFBR2 showed no significant differences between groups. In the RTK/RAS pathway, ERBB4 (3.4% in H/L vs. 1.8% in NHW, p = 0.0369) and ALK (2.7% in H/L vs. 1.1% in NHW, p = 0.0109) mutations were more common in H/L patients. Differences in MET mutation frequency did not reach statistical significance (p = 0.0671). The remaining genes in this pathway showed low mutation rates across both groups. For the WNT pathway, CTNNB1 mutations were significantly more frequent in H/L patients (2.9%) compared to NHW patients (1.3%) (p = 0.01845). AXIN1 mutations were higher in H/L patients (1.0% vs. 0.3%) but not statistically significant (p = 0.07244). Other WNT pathway genes, including APC and RNF43, showed comparable frequencies. PI3K pathway mutations were generally low in both groups, with no statistically significant differences. PIK3CA, PTEN, PIK3R1, AKT2, and MTOR showed slight variability between groups, but none reached significance. In the TP53 pathway, TP53 mutations were found in 68.1% of H/L and 64.7% of NHW patients (p = 0.2015). CDKN2A and ATM mutations were slightly more common in NHW patients (p = 0.06467 and p = 0.06441, respectively), while other genes like MDM2 and CHEK2 had low mutation frequencies across both groups.”
The paragraphs that were updated to the revised Discussion section, line 393-406, now reads:
“Our results suggest pathway-specific molecular differences in PC between H/L and NHW patients. In the TGF-Beta pathway, SMAD2 mutations were more frequent in H/L patients, while SMAD4 mutations were more common in NHW patients, indicating potential differences in tumor suppressor pathway disruption. Higher mutation rates of ERBB4 and ALK in H/L patients suggest ethnicity-specific alterations in the RTK/RAS pathway, possibly linked to differential tumor biology or therapeutic response. CTNNB1 mutations were significantly more frequent in H/L patients, highlighting WNT pathway variability, while overall mutational burden in WNT inhibitors remained low. Although PI3K pathway alterations were infrequent, subtle trends in genes like MTOR and AKT2 may warrant further investigation. TP53 mutation rates were similar between groups, but borderline differences in CDKN2A and ATM suggest variation in DNA repair mechanisms. These results inform the importance of inclusive genomic studies to uncover clinically relevant differences across populations and inform precision medicine strategies tailored to underrepresented groups.”
The paragraphs that were included to the revised Discussion section, lines 393-406, now reads:
“Our results suggest pathway-specific molecular differences in PC between H/L and NHW patients. In the TGF-Beta pathway, SMAD2 mutations were more frequent in H/L patients, while SMAD4 mutations were more common in NHW patients, indicating potential differences in tumor suppressor pathway disruption. Higher mutation rates of ERBB4 and ALK in H/L patients suggest ethnicity-specific alterations in the RTK/RAS pathway, possibly linked to differential tumor biology or therapeutic response. CTNNB1 mutations were significantly more frequent in H/L patients, highlighting WNT pathway variability, while overall mutational burden in WNT inhibitors remained low. Although PI3K pathway alterations were infrequent, subtle trends in genes like MTOR and AKT2 may warrant further investigation. TP53 mutation rates were similar between groups, but borderline differences in CDKN2A and ATM suggest variation in DNA repair mechanisms. These results inform the importance of inclusive genomic studies to uncover clinically relevant differences across populations and inform precision medicine strategies tailored to underrepresented groups.”
Comment 17:
“The discussion mainly contains repetition of the results in too much detailing. Please shorten”.
Response:
We thank the reviewer for this constructive feedback. In response, we have revised and streamlined the Discussion section to reduce repetition and enhance clarity. The updated version emphasizes key interpretative insights while minimizing redundancy with the Results section. Additionally, we have added new paragraphs to address earlier comments.
The updated paragraphs of the Discussion section, lines 364-494, now read:
“…
4.1. Ethnicity-Specific Molecular Differences in PC
Our results reveal distinct molecular profiles in H/L PC patients compared to NHW patients, particularly within the TGF-Beta, RTK/RAS, and WNT pathways. H/L patients had higher SMAD2 mutation rates, while SMAD4 mutations were more common in NHW patients, suggesting possible alternative mechanisms of TGF-Beta pathway dysregulation by ethnicity, potentially involving epigenetic or microenvironmental factors. In the RTK/RAS pathway, ERBB4 and ALK mutations were significantly more frequent in H/L patients, along with enrichment of HRAS and RIT1 mutations—indicating alternative RAS activation routes that may hold therapeutic value beyond KRAS-directed therapies. Similarly, WNT pathway analysis showed a higher prevalence of CTNNB1 mutations in H/L patients, with a trend toward increased AXIN1 mutations. These alterations may contribute to tumor aggressiveness and resistance, highlighting the potential of WNT-targeted therapies in this population.
Our findings highlight notable population-specific differences in pathway-associated survival among pancreatic cancer patients, with particularly stark contrasts observed in the WNT and TP53 pathways. Among H/L patients, WNT alterations were associated with complete loss of survival by the final timepoint, suggesting a potentially aggressive disease course in this group. Conversely, NHW patients with WNT alterations exhibited slightly longer survival, though still poor, underscoring the prognostic importance of this pathway across ancestries. TP53 alterations were also associated with reduced survival in both populations; however, H/L patients without alterations exhibited unexpectedly higher short-term survival, possibly reflecting differences in tumor biology or healthcare access. The PI3K and RTK/RAS pathways showed consistently poor outcomes when altered, with limited data on non-altered comparators, indicating the need for deeper investigation into these oncogenic drivers. Interestingly, TGF-Beta pathway survival was comparable between altered and non-altered groups in NHW patients but more variable among H/L patients. These findings suggest that the prognostic significance of specific pathways may differ by ancestry, reinforcing the importance of population-inclusive genomic profiling and survival analysis to guide precision oncology approaches.
Our results suggest pathway-specific molecular differences in PC between H/L and NHW patients. In the TGF-Beta pathway, SMAD2 mutations were more frequent in H/L patients, while SMAD4 mutations were more common in NHW patients, indicating potential differences in tumor suppressor pathway disruption. Higher mutation rates of ERBB4 and ALK in H/L patients suggest ethnicity-specific alterations in the RTK/RAS pathway, possibly linked to differential tumor biology or therapeutic response. CTNNB1 mutations were significantly more frequent in H/L patients, highlighting WNT pathway variability, while overall mutational burden in WNT inhibitors remained low. Although PI3K pathway alterations were infrequent, subtle trends in genes like MTOR and AKT2 may warrant further investigation. TP53 mutation rates were similar between groups, but borderline differences in CDKN2A and ATM suggest variation in DNA repair mechanisms. These results inform the importance of inclusive genomic studies to uncover clinically relevant differences across populations and inform precision medicine strategies tailored to underrepresented groups.
4.2. Clinical Implications for Risk Prediction and Precision Medicine
Our findings suggest that pathway-specific genomic profiling may improve PC risk prediction and therapeutic strategies across diverse populations. The elevated frequency of SMAD2, ERBB4, ALK, and CTNNB1 mutations in H/L patients may represent potential biomarkers for early detection and inform future treatment strategies in this population. In contrast, the differing patterns of SMAD2 mutations in H/L patients and SMAD4 mutations in NHW patients may reflect underlying ethnicity-specific disease mechanisms
Kaplan-Meier analyses revealed that PI3K and TP53 alterations were significantly associated with poorer survival in NHW patients but not in H/L patients, supporting the potential of PI3K-targeted therapies in the NHW population. The absence of survival associations in H/L patients suggests tumor biology in this group may be shaped by non-genomic factors, such as immune environment, metabolism, or lifestyle risks. Integrating multi-omics approaches will be essential to fully understand these complex influences and inform precision medicine for underrepresented populations.
The observed differences in the nature of mutations across key oncogenic pathways between H/L and NHW patients may have important biological and clinical implications. The predominance of missense mutations in both groups, particularly in genes such as HRAS, ERBB4, CTNNB1, and MET, suggests shared mechanisms of oncogenic activation; however, the mutation spectrum in other genes—such as SMAD4, ATM, and CDKN2A—was notably more diverse in NHW patients. The presence of frame shift, nonsense, and splice site mutations in these genes, especially among NHW cases, may indicate a greater potential for complete loss of function, which could impact tumor suppressor activity and downstream signaling pathways. In contrast, the higher proportion of missense mutations in H/L patients, particularly within SMAD2 and CTNNB1, may result in more nuanced functional alterations that contribute to tumor progression through different biological mechanisms. Additionally, the absence of translation start site mutations in the H/L cohort highlights potential ethnic differences in mutational patterns that warrant further functional validation. These findings inform the importance of not only identifying which genes are mutated, but also characterizing the type of mutation, as this can influence both prognostic interpretation and therapeutic strategies in a population-specific context.
4.3. Limitations and Future Directions
Although this study offers one of the most comprehensive ethnicity-specific genomic analyses of PC to date, it has limitations. The retrospective design and underrepresentation of H/L patients may introduce selection bias and limit statistical power, particularly for rare mutations. Future validation in larger, prospectively collected, ethnically balanced cohorts is needed. Additionally, functional studies—such as patient-derived organoids and single-cell sequencing—are essential to elucidate the biological impact of SMAD2, ERBB4, ALK, and CTNNB1 mutations and their roles in tumor heterogeneity, drug resistance, and immune evasion.
While H/L individuals have a lower overall incidence of PC compared to NHW patients, they are disproportionately diagnosed at younger ages and often with more advanced disease. This apparent paradox suggests that incidence alone may not fully capture the clinical burden of PC in this population. Contributing factors may include delayed access to care, lower screening rates, or environmental and lifestyle risks such as high rates of diabetes and obesity that could accelerate tumor progression in H/L patients. Additionally, the younger age at diagnosis may reflect an underlying biological susceptibility that is not yet fully understood, possibly related to distinct genomic or epigenomic profiles. Our findings of specific pathway alterations, including enriched mutations in SMAD2, ERBB4, ALK, and CTNNB1 among H/L patients, support the hypothesis that early-onset PC in this population may follow a unique molecular trajectory. These observations reinforce the importance of investigating age- and ethnicity-specific molecular drivers to inform early detection strategies and tailored treatment approaches.
A limitation of this study is the absence of tumor stage and grade information in the dataset analyzed. While these are important prognosticators in PC, the dataset only included site classification (primary vs. metastatic). This limitation restricts our ability to assess how molecular alterations may correlate with traditional clinicopathological parameters. Nonetheless, this dataset remains one of the few publicly available resources enabling population-specific genomic analyses, particularly among underrepresented groups.
A constraint of this study is the lack of granular histopathological data within the publicly available dataset used for analysis. Specifically, while we stratified cases based on OncoTree classification and identified pancreatic adenocarcinoma as the predominant diagnosis, the database did not include detailed histological subtyping to distinguish between ductal and acinar forms. Although we were able to report specific cancer types such as adenocarcinoma and acinar cell carcinoma, the inability to confirm ductal versus acinar histology for all cases limits our capacity to perform subtype-specific analyses. This inform the need for more comprehensive clinical annotations in public genomic datasets to enhance the precision and applicability of molecular findings.
Another limitation of this analysis is the imbalance in sample sizes between H/L and NHW patients, with 407 H/L cases compared to 3,841 NHW cases. This discrepancy reflects the current underrepresentation of H/L populations in publicly available PC genomic datasets. While the statistical methods employed, such as chi-squared tests, are appropriate and robust for detecting differences even in unbalanced cohorts, the disparity in sample size may still affect the sensitivity and generalizability of some comparisons. This limitation highlights a critical gap in cancer genomics research and reinforces the need for greater inclusion of diverse populations in future genomic studies to ensure that molecular insights and precision medicine strategies are equitably informed across all patient groups.
An important consideration in this study is the lack of control for clinical and sociodemographic confounders, such as tumor stage, treatment type, comorbidities, and socioeconomic status. These variables were inconsistently reported or unavailable across the analyzed datasets, which may impact both genomic profiles and survival outcomes. As such, our findings should be viewed as exploratory and hypothesis-generating rather than conclusive. The differences observed between H/L and NHW patients may, in part, be influenced by unmeasured clinical or contextual factors. Moving forward, studies that integrate comprehensive clinical and demographic data alongside genomic information will be critical to more precisely define the molecular contributors to PC disparities and to support equitable implementation of precision oncology strategies.
….”
We thank the reviewer for encouraging us to highlight this critical need and ongoing efforts that will enhance the future impact of this work.

Reviewer 2 Report
Comments and Suggestions for Authors
This manuscript explores ethnicity-specific genomic alterations in pancreatic cancer using publicly available datasets (TCGA and GENIE). The analysis focuses on pathway-level mutation differences and their association with clinical outcomes, a relevant and vital topic in cancer disparities research. However, several key concerns remain that limit the current impact and interpretability of the findings:
1. Unclear Novelty and Study Rationale: While the topic is relevant, the manuscript does not clearly articulate how it adds new insight beyond prior studies on racial disparities in pancreatic cancer genomics. The introduction should be revised to better justify the focus on pathway-level mutation profiling by race/ethnicity and clarify what distinguishes this analysis from existing literature.
2. Overinterpretation of Descriptive Results: The conclusions overstate the significance of the findings. As this is an observational study based entirely on retrospective genomic data without functional validation or treatment control, any findings regarding therapeutic relevance or outcome prediction should be framed as exploratory. Phrases like “could guide personalized therapy” should be softened accordingly.
3. Lack of Confounder Control: Important clinical variables such as tumor stage, treatment type, and socioeconomic status are not addressed or controlled for in the analysis. These factors could significantly influence both mutation profiles and outcomes. At a minimum, these limitations should be explicitly discussed.
4. Inconsistent Sample Sizes and Statistical Power: Some subgroup analyses, particularly survival comparisons by race and pathway status, appear underpowered. Confidence intervals and effect sizes should be reported to provide context for the robustness of these findings.
5. Language and Clarity: The manuscript contains long, overly technical sentences, redundant phrasing, and occasional grammatical errors. A professional English editing service is recommended to improve clarity and flow, particularly in the introduction and discussion.
6. Supplementary Material Transparency: Table S1 contains important annotation details that should be referenced more explicitly in the main text. Consider directly integrating key pathway mutation definitions into the Methods or Results for greater accessibility.
The manuscript is generally understandable, but the quality of English requires improvement. Several sections contain long, overly complex sentences, awkward phrasing, and redundant expressions. These issues affect clarity and may hinder reader comprehension. A thorough professional language edit is strongly recommended to improve sentence structure, grammar, and overall readability. Refining the language will enhance the manuscript’s impact and ensure the scientific content is communicated more effectively.
Author Response
Responses to Reviewer 2 Comments are in the Word file: Response_Reviewer_2_Comments_042325_EV.docx
Reviewer 2 Comments
We are pleased to re-submit this manuscript and believe it offers important insights that will resonate with the scientific and clinical research community. We have thoroughly addressed all reviewer comments and refined the presentation to underscore the significance of this cancer disparity topic. Our study presents a comprehensive, pathway-level analysis of pancreatic cancer using one of the few publicly available genomic datasets with sufficient representation of Hispanic/Latino (H/L) patients. This work represents one of the first ethnicity-focused investigations of pathway-specific genomic alterations in pancreatic cancer, with particular attention to key drivers such as TGF-Beta, RTK/RAS, WNT, PI3K, and TP53. Notably, we identify SMAD2, ERBB4, ALK, and CTNNB1 as recurrently altered genes in H/L patients, emphasizing the importance of ethnicity-specific molecular profiling for improving risk stratification, early detection, and targeted treatment strategies in underrepresented populations.
Thank you very much for taking the time to review this manuscript. Please find the detailed responses below and the corresponding revisions wrote in blue font and highlighted in yellow in the re-submitted Word file.
Reviewer 2’s feedback was positive. Reviewer 2 provided thoughtful and constructive feedback, acknowledging the relevance of our manuscript titled “Pathway-specific genomic alterations in pancreatic cancer across populations at risk” and its focus on ethnicity-specific genomic differences. The reviewer recognized the importance of exploring pathway-level mutation differences in the context of cancer disparities but raised several critical points that have guided meaningful revisions to enhance the manuscript’s clarity, rigor, and interpretability. In response, we revised the Introduction to better articulate the novelty and rationale of our study, emphasizing how our pathway-based, ethnicity-stratified approach contributes new insights beyond prior work. We also refined the language in the Conclusions to frame therapeutic implications as exploratory, consistent with the retrospective nature of the data. To address limitations, we expanded the Discussion to acknowledge the lack of clinical confounder control and underpowered subgroup analyses, and we now report confidence intervals to contextualize effect sizes. Furthermore, we improved the overall readability of the manuscript by simplifying technical language and reducing redundancy, and we clarified the use and referencing of Supplementary Table S1 by integrating key pathway mutation definitions into the Methods section. We sincerely thank Reviewer 2 for their detailed critique, which has significantly strengthened the manuscript.
Reviewer 2 writes:
“This manuscript explores ethnicity-specific genomic alterations in pancreatic cancer using publicly available datasets (TCGA and GENIE). The analysis focuses on pathway-level mutation differences and their association with clinical outcomes, a relevant and vital topic in cancer disparities research. However, several key concerns remain that limit the current impact and interpretability of the findings.”
We thank Reviewer 2 for their constructive and encouraging feedback. We are pleased to hear that the reviewer recognized the relevance of our manuscript and the importance of exploring ethnicity-specific genomic alterations in pancreatic cancer using large, publicly available datasets. We appreciate the acknowledgment that our focus on pathway-level mutation differences and their association with clinical outcomes addresses a vital topic in cancer disparities research. In response to the reviewer’s comments, we have carefully revised the manuscript to clarify the study’s novelty, strengthen the interpretation of findings, and improve the overall clarity and impact of the work.
Comment 1:
- “Unclear Novelty and Study Rationale: While the topic is relevant, the manuscript does not clearly articulate how it adds new insight beyond prior studies on racial disparities in pancreatic cancer genomics. The introduction should be revised to better justify the focus on pathway-level mutation profiling by race/ethnicity and clarify what distinguishes this analysis from existing literature”.
Response:
We thank Reviewer 2 for this important comment regarding the novelty and rationale of our study. In response, we have revised the Introduction to more clearly articulate how our pathway-level approach adds value beyond existing literature on racial disparities in pancreatic cancer. Specifically, we emphasize that while prior studies have broadly characterized genomic differences by race/ethnicity, our study uniquely focuses on the distribution and clinical relevance of mutations within five key oncogenic pathways—TGF-Beta, RTK/RAS, WNT, PI3K, and TP53—stratified by ethnicity.
To further strengthen the originality and interpretability of our findings, we have included a new Supplementary Table S2, which details the nature and distribution of gene mutation types within these pathways in both Hispanic/Latino (H/L) and Non-Hispanic White (NHW) patients. We also added a dedicated paragraph in the Results section describing these mutation patterns, and an expanded discussion in the Discussion section that explores their potential biological implications and relevance across populations.
Additionally, we have moved the OncoTree code information from Table 2 to Table 1 for improved clarity, and we revised the classification of tumor types accordingly. We acknowledge the reviewer’s point regarding the inappropriate inclusion of IPMN and MCN under pancreatic cancer, and we have updated the terminology throughout the manuscript to more accurately refer to these cases as “pancreatic tumors,” encompassing all neoplastic entities, both benign and malignant. We have also reviewed and clarified the histological subtypes used in our dataset to ensure appropriate inclusion criteria for the cohort. These changes are now reflected in the revised version of the manuscript.
The revised text on the Abstract section, lines 20-37 now reads: “Abstract: Background/Objectives:Pancreatic cancer (PC) is a highly aggressive malignancy with increasing incidence and poor survival. Hispanic/Latino (H/L) patients, despite having a lower overall incidence than Non-Hispanic White (NHW) patients, are often diagnosed younger and at more advanced stages, leading to worse outcomes. The molecular mechanisms underlying these disparities remain unclear. This study characterizes mutations in key oncogenic pathways—TP53, WNT, PI3K, TGF-Beta, and RTK/RAS—among H/L and NHW patients using publicly available datasets. Methods: We analyzed genomic data from 4,248 PC patients (407 H/L; 3,841 NHW), comparing mutation frequencies across pathways. Chi-squared tests assessed group differences, and Kaplan-Meier analysis evaluated survival outcomes by pathway alterations. Results: TGF-Beta pathway mutations were less common in H/L patients (18.4% vs. 24.4%, p = 8.6e-3), with notable differences in SMAD2 (1.5% vs. 0.4%, p = 6.3e-3) and SMAD4 (15% vs. 19.9%, p = 0.02). While overall differences in other pathways were not statistically significant, several genes showed borderline significance, including ERBB4, ALK, HRAS, RIT1 (RTK/RAS), and CTNNB1 (WNT). No significant survival differences were observed in H/L patients, but NHW patients with TP53 alterations showed borderline survival associations. Conclusions: This study reveals ethnicity-specific pathway alterations in PC, with SMAD2, ERBB4, ALK, and CTNNB1 mutations more frequent in H/L patients, while SMAD4 and PI3K alterations had prognostic value in NHW patients. These findings underscore the importance of incorporating ethnicity-specific molecular profiling into precision oncology for pancreatic cancer.”
The new Table S2 in the supplementary material, now reads:
“Table S2. Nature of gene mutations within the namely TP53, WNT, PI3K, TGF-Beta, and RTK/RAS pathways in pancreatic cancer (PC) among Hispanic/Latino (H/L) and non-Hispanic White (NHW) patients. Mutation types include frame shift deletions, frame shift insertions, missense mutations, nonsense mutations, splice site mutations, and translation start site mutations.”
The paragraph that were updated to the revised Methos section, lines 0 – 1, now reads:
“To complement our analysis of mutation frequencies, we examined the types of genomic alterations in key genes within the TP53, WNT, PI3K, TGF-Beta, and RTK/RAS pathways among H/L and NHW PC patients. Mutation types included frame shift, in-frame, missense, nonsense, splice site, and translation start site mutations, as detailed in Supplementary Table S2. Missense mutations were the most common across both groups, particularly in HRAS, MET, CTNNB1, ERBB4, and RIT1. In contrast, genes like SMAD4, ATM, and CDKN2A showed more diverse mutation profiles, including a notable proportion of truncating and splice site variants. These data provide additional insight into potential functional differences in mutation patterns across populations.”
The paragraphs that were updated and added to the revised Results section, line 260-345, now reads:
“Kaplan-Meier survival analysis for TGF-Beta pathway alterations in NHW PC patients showed no statistically significant difference in OS between altered and non-altered groups (p = 0.29; Figure S1). The closely aligned survival curves and broad confidence intervals suggest limited prognostic value of TGF-Beta pathway mutations in this cohort, though variability in survival outcomes indicates underlying heterogeneity that warrants further investigation in larger, more diverse populations. For the RTK/RAS pathway, patients with alterations showed a trend toward reduced survival compared to those without, but this was not statistically significant (p = 0.61; Figure S1). The survival curve for the altered group declined more gradually, and broad confidence intervals reflect high variability. These findings suggest that RTK/RAS pathway alterations may not have a strong prognostic impact in NHW PC patients, although larger cohorts may help reveal subgroup-specific effects. Similarly, WNT pathway alterations were not associated with significant OS differences (p = 0.65; Figure S1). Survival curves for both groups were nearly identical, with overlapping confidence intervals indicating minimal distinction in outcomes. Despite the biological relevance of WNT signaling, these results suggest limited prognostic value in this population, though further research with larger datasets may help clarify potential roles. In contrast, PI3K pathway alterations were significantly associated with poorer OS (p = 0.019; Figure S1). Patients with alterations showed a more pronounced decline in survival, with clear separation between survival curves. While confidence intervals were wide, these findings suggest PI3K pathway dysregulation may contribute to disease progression and could serve as a therapeutic target in NHW PC. Likewise, TP53 pathway alterations were linked to significantly worse OS (p = 0.0472; Figure S1). Patients with TP53 mutations had a marked decline in survival compared to those without, suggesting a critical role in disease progression. Despite variability in the altered group, these results support the prognostic relevance of TP53 alterations and highlight their potential for informing precision medicine approaches.
Overall, these results reveal pathway-specific differences in survival among NHW PC patients. While TGF-Beta, RTK/RAS, and WNT pathways were not significantly associated with survival outcomes, PI3K and TP53 pathway alterations were linked to poorer prognosis. These findings inform the importance of comprehensive genomic profiling to inform risk stratification and guide targeted therapies, and they highlight the need for larger, well-annotated cohorts to validate and expand upon these insights.
Survival differences across key oncogenic pathways varied by ancestry among pancreatic cancer patients. In the H/L cohort, WNT pathway alterations were associated with a survival probability of 0.00 at 71.7 months, compared to 0.39 in patients without alterations at 24.5 months (95% CI: 0.20–0.74). For TGF-Beta pathway alterations, survival was 0.30 (95% CI: 0.06–1.00) among altered cases versus 0.34 (95% CI: 0.14–0.81) in non-altered cases. TP53 alterations in H/L patients were associated with a 0.29 survival probability at 71.7 months (95% CI: 0.10–0.78), while non-altered patients had a survival of 0.50 at 7.4 months (95% CI: 0.13–1.00). For the PI3K pathway, survival for non-altered H/L patients was 0.30 at 71.7 months (95% CI: 0.11–0.81), while no long-term survival data were available for the altered group. Similarly, RTK/RAS pathway alterations yielded a survival of 0.27 at 71.7 months (95% CI: 0.10–0.73), with no non-altered group data at that timepoint. Among NHW patients, survival differences were also observed. WNT pathway alterations were linked to a 0.11 survival probability at 87.1 months (95% CI: 0.04–0.33), compared to 0.18 in non-altered patients at 80.9 months (95% CI: 0.14–0.23). For TGF-Beta, survival was 0.17 for both altered (69.9 months, 95% CI: 0.11–0.26) and non-altered groups (87.1 months, 95% CI: 0.13–0.23). TP53-altered patients had a survival of 0.16 at 87.1 months (95% CI: 0.12–0.21), whereas non-altered patients had 0.22 survival at 80.9 months (95% CI: 0.14–0.34). PI3K-altered patients had a survival of 0.09 at 62.8 months (95% CI: 0.02–0.49), while non-altered patients had 0.18 survival at 87.1 months (95% CI: 0.14–0.23). RTK/RAS pathway alterations were associated with a survival of 0.27 at 73.7 months (95% CI: 0.14–0.41), with no final timepoint data available for the non-altered group.
The analysis of pathway-specific gene mutations among H/L and NHW PC patients revealed several differences in mutation frequencies (Supplementary Table S1). Within the TGF-Beta pathway, SMAD2 mutations were more frequent in H/L patients (1.5%) compared to NHW patients (0.4%) (p = 0.0064), while SMAD4 mutations were more common in NHW patients (19.9%) versus H/L patients (15.0%) (p = 0.0202). Other genes such as SMAD3, TGFBR1, and TGFBR2 showed no significant differences between groups. In the RTK/RAS pathway, ERBB4 (3.4% in H/L vs. 1.8% in NHW, p = 0.0369) and ALK (2.7% in H/L vs. 1.1% in NHW, p = 0.0109) mutations were more common in H/L patients. Differences in MET mutation frequency did not reach statistical significance (p = 0.0671). The remaining genes in this pathway showed low mutation rates across both groups. For the WNT pathway, CTNNB1 mutations were significantly more frequent in H/L patients (2.9%) compared to NHW patients (1.3%) (p = 0.01845). AXIN1 mutations were higher in H/L patients (1.0% vs. 0.3%) but not statistically significant (p = 0.07244). Other WNT pathway genes, including APC and RNF43, showed comparable frequencies. PI3K pathway mutations were generally low in both groups, with no statistically significant differences. PIK3CA, PTEN, PIK3R1, AKT2, and MTOR showed slight variability between groups, but none reached significance. In the TP53 pathway, TP53 mutations were found in 68.1% of H/L and 64.7% of NHW patients (p = 0.2015). CDKN2A and ATM mutations were slightly more common in NHW patients (p = 0.06467 and p = 0.06441, respectively), while other genes like MDM2 and CHEK2 had low mutation frequencies across both groups.
To further explore the nature of pathway-specific genomic alterations, we analyzed the types of mutations present in key genes across the TP53, WNT, PI3K, TGF-Beta, and RTK/RAS pathways among H/L and NH) patients (Supplementary Table S2). Missense mutations were the most common alteration type across both populations, particularly in genes such as HRAS, ERBB4, CTNNB1, MET, and RIT1, where they accounted for over 75% of observed mutations in both cohorts. In contrast, genes like SMAD4 and CDKN2A displayed a more diverse mutation profile, including a substantial proportion of frame shift deletions, nonsense mutations, and splice site alterations. Notably, H/L patients showed a higher proportion of missense mutations in SMAD2 (66.7%) and CTNNB1 (91.7%), while NHW patients exhibited a broader distribution of mutation types in SMAD4 and ATM, including translation start site mutations, which were not observed in the H/L cohort. These differences in mutation type and frequency may have implications for protein function, tumor behavior, and response to targeted therapies.”
The paragraphs that were updated to the revised Results section, lines 313 - 332, now reads:
“The analysis of pathway-specific gene mutations among H/L and NHW PC patients revealed several differences in mutation frequencies (Supplementary Table S1). Within the TGF-Beta pathway, SMAD2 mutations were more frequent in H/L patients (1.5%) compared to NHW patients (0.4%) (p = 0.0064), while SMAD4 mutations were more common in NHW patients (19.9%) versus H/L patients (15.0%) (p = 0.0202). Other genes such as SMAD3, TGFBR1, and TGFBR2 showed no significant differences between groups. In the RTK/RAS pathway, ERBB4 (3.4% in H/L vs. 1.8% in NHW, p = 0.0369) and ALK (2.7% in H/L vs. 1.1% in NHW, p = 0.0109) mutations were more common in H/L patients. Differences in MET mutation frequency did not reach statistical significance (p = 0.0671). The remaining genes in this pathway showed low mutation rates across both groups. For the WNT pathway, CTNNB1 mutations were significantly more frequent in H/L patients (2.9%) compared to NHW patients (1.3%) (p = 0.01845). AXIN1 mutations were higher in H/L patients (1.0% vs. 0.3%) but not statistically significant (p = 0.07244). Other WNT pathway genes, including APC and RNF43, showed comparable frequencies. PI3K pathway mutations were generally low in both groups, with no statistically significant differences. PIK3CA, PTEN, PIK3R1, AKT2, and MTOR showed slight variability between groups, but none reached significance. In the TP53 pathway, TP53 mutations were found in 68.1% of H/L and 64.7% of NHW patients (p = 0.2015). CDKN2A and ATM mutations were slightly more common in NHW patients (p = 0.06467 and p = 0.06441, respectively), while other genes like MDM2 and CHEK2 had low mutation frequencies across both groups.”
The paragraphs that were updated to the revised Discussion section, line 393-406, now reads:
“Our results suggest pathway-specific molecular differences in PC between H/L and NHW patients. In the TGF-Beta pathway, SMAD2 mutations were more frequent in H/L patients, while SMAD4 mutations were more common in NHW patients, indicating potential differences in tumor suppressor pathway disruption. Higher mutation rates of ERBB4 and ALK in H/L patients suggest ethnicity-specific alterations in the RTK/RAS pathway, possibly linked to differential tumor biology or therapeutic response. CTNNB1 mutations were significantly more frequent in H/L patients, highlighting WNT pathway variability, while overall mutational burden in WNT inhibitors remained low. Although PI3K pathway alterations were infrequent, subtle trends in genes like MTOR and AKT2 may warrant further investigation. TP53 mutation rates were similar between groups, but borderline differences in CDKN2A and ATM suggest variation in DNA repair mechanisms. These results inform the importance of inclusive genomic studies to uncover clinically relevant differences across populations and inform precision medicine strategies tailored to underrepresented groups.”
The paragraphs that were included to the revised Discussion section, lines 393-406, now reads:
“Our results suggest pathway-specific molecular differences in PC between H/L and NHW patients. In the TGF-Beta pathway, SMAD2 mutations were more frequent in H/L patients, while SMAD4 mutations were more common in NHW patients, indicating potential differences in tumor suppressor pathway disruption. Higher mutation rates of ERBB4 and ALK in H/L patients suggest ethnicity-specific alterations in the RTK/RAS pathway, possibly linked to differential tumor biology or therapeutic response. CTNNB1 mutations were significantly more frequent in H/L patients, highlighting WNT pathway variability, while overall mutational burden in WNT inhibitors remained low. Although PI3K pathway alterations were infrequent, subtle trends in genes like MTOR and AKT2 may warrant further investigation. TP53 mutation rates were similar between groups, but borderline differences in CDKN2A and ATM suggest variation in DNA repair mechanisms. These results inform the importance of inclusive genomic studies to uncover clinically relevant differences across populations and inform precision medicine strategies tailored to underrepresented groups.”
The updated paragraphs of the Discussion section, lines 364-494, now read:
“…
4.1. Ethnicity-Specific Molecular Differences in PC
Our results reveal distinct molecular profiles in H/L PC patients compared to NHW patients, particularly within the TGF-Beta, RTK/RAS, and WNT pathways. H/L patients had higher SMAD2 mutation rates, while SMAD4 mutations were more common in NHW patients, suggesting possible alternative mechanisms of TGF-Beta pathway dysregulation by ethnicity, potentially involving epigenetic or microenvironmental factors. In the RTK/RAS pathway, ERBB4 and ALK mutations were significantly more frequent in H/L patients, along with enrichment of HRAS and RIT1 mutations—indicating alternative RAS activation routes that may hold therapeutic value beyond KRAS-directed therapies. Similarly, WNT pathway analysis showed a higher prevalence of CTNNB1 mutations in H/L patients, with a trend toward increased AXIN1 mutations. These alterations may contribute to tumor aggressiveness and resistance, highlighting the potential of WNT-targeted therapies in this population.
Our findings highlight notable population-specific differences in pathway-associated survival among pancreatic cancer patients, with particularly stark contrasts observed in the WNT and TP53 pathways. Among H/L patients, WNT alterations were associated with complete loss of survival by the final timepoint, suggesting a potentially aggressive disease course in this group. Conversely, NHW patients with WNT alterations exhibited slightly longer survival, though still poor, underscoring the prognostic importance of this pathway across ancestries. TP53 alterations were also associated with reduced survival in both populations; however, H/L patients without alterations exhibited unexpectedly higher short-term survival, possibly reflecting differences in tumor biology or healthcare access. The PI3K and RTK/RAS pathways showed consistently poor outcomes when altered, with limited data on non-altered comparators, indicating the need for deeper investigation into these oncogenic drivers. Interestingly, TGF-Beta pathway survival was comparable between altered and non-altered groups in NHW patients but more variable among H/L patients. These findings suggest that the prognostic significance of specific pathways may differ by ancestry, reinforcing the importance of population-inclusive genomic profiling and survival analysis to guide precision oncology approaches.
Our results suggest pathway-specific molecular differences in PC between H/L and NHW patients. In the TGF-Beta pathway, SMAD2 mutations were more frequent in H/L patients, while SMAD4 mutations were more common in NHW patients, indicating potential differences in tumor suppressor pathway disruption. Higher mutation rates of ERBB4 and ALK in H/L patients suggest ethnicity-specific alterations in the RTK/RAS pathway, possibly linked to differential tumor biology or therapeutic response. CTNNB1 mutations were significantly more frequent in H/L patients, highlighting WNT pathway variability, while overall mutational burden in WNT inhibitors remained low. Although PI3K pathway alterations were infrequent, subtle trends in genes like MTOR and AKT2 may warrant further investigation. TP53 mutation rates were similar between groups, but borderline differences in CDKN2A and ATM suggest variation in DNA repair mechanisms. These results inform the importance of inclusive genomic studies to uncover clinically relevant differences across populations and inform precision medicine strategies tailored to underrepresented groups.
4.2. Clinical Implications for Risk Prediction and Precision Medicine
Our findings suggest that pathway-specific genomic profiling may improve PC risk prediction and therapeutic strategies across diverse populations. The elevated frequency of SMAD2, ERBB4, ALK, and CTNNB1 mutations in H/L patients may represent potential biomarkers for early detection and inform future treatment strategies in this population. In contrast, the differing patterns of SMAD2 mutations in H/L patients and SMAD4 mutations in NHW patients may reflect underlying ethnicity-specific disease mechanisms
Kaplan-Meier analyses revealed that PI3K and TP53 alterations were significantly associated with poorer survival in NHW patients but not in H/L patients, supporting the potential of PI3K-targeted therapies in the NHW population. The absence of survival associations in H/L patients suggests tumor biology in this group may be shaped by non-genomic factors, such as immune environment, metabolism, or lifestyle risks. Integrating multi-omics approaches will be essential to fully understand these complex influences and inform precision medicine for underrepresented populations.
The observed differences in the nature of mutations across key oncogenic pathways between H/L and NHW patients may have important biological and clinical implications. The predominance of missense mutations in both groups, particularly in genes such as HRAS, ERBB4, CTNNB1, and MET, suggests shared mechanisms of oncogenic activation; however, the mutation spectrum in other genes—such as SMAD4, ATM, and CDKN2A—was notably more diverse in NHW patients. The presence of frame shift, nonsense, and splice site mutations in these genes, especially among NHW cases, may indicate a greater potential for complete loss of function, which could impact tumor suppressor activity and downstream signaling pathways. In contrast, the higher proportion of missense mutations in H/L patients, particularly within SMAD2 and CTNNB1, may result in more nuanced functional alterations that contribute to tumor progression through different biological mechanisms. Additionally, the absence of translation start site mutations in the H/L cohort highlights potential ethnic differences in mutational patterns that warrant further functional validation. These findings inform the importance of not only identifying which genes are mutated, but also characterizing the type of mutation, as this can influence both prognostic interpretation and therapeutic strategies in a population-specific context.
4.3. Limitations and Future Directions
Although this study offers one of the most comprehensive ethnicity-specific genomic analyses of PC to date, it has limitations. The retrospective design and underrepresentation of H/L patients may introduce selection bias and limit statistical power, particularly for rare mutations. Future validation in larger, prospectively collected, ethnically balanced cohorts is needed. Additionally, functional studies—such as patient-derived organoids and single-cell sequencing—are essential to elucidate the biological impact of SMAD2, ERBB4, ALK, and CTNNB1 mutations and their roles in tumor heterogeneity, drug resistance, and immune evasion.
While H/L individuals have a lower overall incidence of PC compared to NHW patients, they are disproportionately diagnosed at younger ages and often with more advanced disease. This apparent paradox suggests that incidence alone may not fully capture the clinical burden of PC in this population. Contributing factors may include delayed access to care, lower screening rates, or environmental and lifestyle risks such as high rates of diabetes and obesity that could accelerate tumor progression in H/L patients. Additionally, the younger age at diagnosis may reflect an underlying biological susceptibility that is not yet fully understood, possibly related to distinct genomic or epigenomic profiles. Our findings of specific pathway alterations, including enriched mutations in SMAD2, ERBB4, ALK, and CTNNB1 among H/L patients, support the hypothesis that early-onset PC in this population may follow a unique molecular trajectory. These observations reinforce the importance of investigating age- and ethnicity-specific molecular drivers to inform early detection strategies and tailored treatment approaches.
A limitation of this study is the absence of tumor stage and grade information in the dataset analyzed. While these are important prognosticators in PC, the dataset only included site classification (primary vs. metastatic). This limitation restricts our ability to assess how molecular alterations may correlate with traditional clinicopathological parameters. Nonetheless, this dataset remains one of the few publicly available resources enabling population-specific genomic analyses, particularly among underrepresented groups.
A constraint of this study is the lack of granular histopathological data within the publicly available dataset used for analysis. Specifically, while we stratified cases based on OncoTree classification and identified pancreatic adenocarcinoma as the predominant diagnosis, the database did not include detailed histological subtyping to distinguish between ductal and acinar forms. Although we were able to report specific cancer types such as adenocarcinoma and acinar cell carcinoma, the inability to confirm ductal versus acinar histology for all cases limits our capacity to perform subtype-specific analyses. This inform the need for more comprehensive clinical annotations in public genomic datasets to enhance the precision and applicability of molecular findings.
Another limitation of this analysis is the imbalance in sample sizes between H/L and NHW patients, with 407 H/L cases compared to 3,841 NHW cases. This discrepancy reflects the current underrepresentation of H/L populations in publicly available PC genomic datasets. While the statistical methods employed, such as chi-squared tests, are appropriate and robust for detecting differences even in unbalanced cohorts, the disparity in sample size may still affect the sensitivity and generalizability of some comparisons. This limitation highlights a critical gap in cancer genomics research and reinforces the need for greater inclusion of diverse populations in future genomic studies to ensure that molecular insights and precision medicine strategies are equitably informed across all patient groups.
An important consideration in this study is the lack of control for clinical and sociodemographic confounders, such as tumor stage, treatment type, comorbidities, and socioeconomic status. These variables were inconsistently reported or unavailable across the analyzed datasets, which may impact both genomic profiles and survival outcomes. As such, our findings should be viewed as exploratory and hypothesis-generating rather than conclusive. The differences observed between H/L and NHW patients may, in part, be influenced by unmeasured clinical or contextual factors. Moving forward, studies that integrate comprehensive clinical and demographic data alongside genomic information will be critical to more precisely define the molecular contributors to PC disparities and to support equitable implementation of precision oncology strategies.
….”
Comment 2:
- “Overinterpretation of Descriptive Results: The conclusions overstate the significance of the findings. As this is an observational study based entirely on retrospective genomic data without functional validation or treatment control, any findings regarding therapeutic relevance or outcome prediction should be framed as exploratory. Phrases like “could guide personalized therapy” should be softened accordingly.”
Response:
We thank Reviewer 2 for this insightful comment. We agree that, as an observational study based on retrospective genomic data without functional validation or treatment control, our findings should be interpreted with appropriate caution. In response, we have carefully revised the language throughout the Results, Discussion, and Conclusion sections to avoid overinterpretation and to more accurately reflect the exploratory nature of our analyses. Specifically, we have softened statements regarding clinical implications and therapeutic relevance—for example, replacing phrases such as “could guide personalized therapy” with more measured language such as “may inform future investigations”. These revisions ensure that the conclusions align with the scope and limitations of the study, and we thank the reviewer for helping us improve the balance and scientific rigor of our manuscript.
The updated paragraphs of the Results section, lines 196-198, now read:
“These findings highlight ethnicity-specific molecular variations in PC, suggesting that H/L patients exhibit distinct genomic profiles that may influence tumor progression and potential treatment strategies.”
The updated paragraphs of the Results section, lines 347-350, now read:
“However, the lack of statistical significance in most comparisons emphasizes the need for larger studies to validate these observations and explore their potential implications for precision medicine and targeted therapy in H/L PC patients.”
The updated paragraphs of the Conclusions section, lines 409-413, now read:
“…The elevated frequency of SMAD2, ERBB4, ALK, and CTNNB1 mutations in H/L patients may representpotential biomarkers for early detection and inform future treatment strategies in this population. In contrast, the differing patterns of SMAD2 mutations in H/L patients and SMAD4 mutations in NHW patients may reflect underlying ethnicity-specific disease mechanisms.”
The updated paragraphs of the Conclusions section, lines 504-506, now read:
“These findings may inform future investigations of ethnicity-specific molecular profiling for potential risk stratification, early detection, and targeted treatment selection.
Comment 3:
- “Lack of Confounder Control: Important clinical variables such as tumor stage, treatment type, and socioeconomic status are not addressed or controlled for in the analysis. These factors could significantly influence both mutation profiles and outcomes. At a minimum, these limitations should be explicitly discussed”.
Response:
We thank Reviewer 2 for this important observation. We fully acknowledge that clinical variables such as tumor stage, treatment type, and socioeconomic status may influence both mutation profiles and survival outcomes. Unfortunately, these variables were not consistently available across the publicly accessible datasets used in our analysis (TCGA and GENIE), limiting our ability to adjust for potential confounders in a statistically robust manner. To address this limitation, we have added a dedicated paragraph in the Discussion section explicitly acknowledging the absence of confounder control and its implications for interpreting our findings. While our study provides one of the most comprehensive analyses of pathway-specific genomic alterations in H/L versus NHW pancreatic cancer patients to date, we now emphasize that our conclusions are exploratory in nature and should be interpreted within the context of these limitations. We also highlight the need for future studies with integrated clinical-genomic data and balanced representation across racial/ethnic groups to more definitively evaluate the relationship between molecular alterations, clinical factors, and outcomes.
The added paragraph of the Discussion section, lines 485-494, now read:
“An important consideration in this study is the lack of control for clinical and sociodemographic confounders, such as tumor stage, treatment type, comorbidities, and socioeconomic status. These variables were inconsistently reported or unavailable across the analyzed datasets, which may impact both genomic profiles and survival outcomes. As such, our findings should be viewed as exploratory and hypothesis-generating rather than conclusive. The differences observed between H/L and NHW patients may, in part, be influenced by unmeasured clinical or contextual factors. Moving forward, studies that integrate comprehensive clinical and demographic data alongside genomic information will be critical to more precisely define the molecular contributors to pancreatic cancer disparities and to support equitable implementation of precision oncology strategies.”
Comment 4:
“Inconsistent Sample Sizes and Statistical Power: Some subgroup analyses, particularly survival comparisons by race and pathway status, appear underpowered. Confidence intervals and effect sizes should be reported to provide context for the robustness of these findings”.
Response:
We thank Reviewer 2 for this important observation regarding sample size and statistical power. We acknowledge that the survival analyses, particularly those stratified by race and pathway alteration status, may be underpowered due to the relatively small number of Hispanic/Latino (H/L) patients included in the publicly available datasets. As noted in the Materials and Methods and Results sections, our H/L cohort consisted of 407 patients, which, although one of the largest assembled to date for this population, remains limited in power for subgroup analyses—especially for low-frequency mutations. In response to this comment, we have revised the manuscript to include two new paragraphs for the Results and Discussions about the confidence intervals and effect sizes in all survival analyses to provide greater transparency regarding the robustness of these estimates. Additionally, we have explicitly discussed these limitations in the Discussion section, emphasizing that the findings should be interpreted as exploratory and hypothesis-generating. We also underscore the urgent need for larger, well-annotated datasets with balanced racial and ethnic representation to validate these preliminary findings and to more confidently assess the prognostic significance of pathway-specific genomic alterations in underrepresented populations. We appreciate the reviewer’s suggestion, which has helped us improve the clarity and scientific rigor of our manuscript.
The added paragraph of the Results section, lines 292-312, now read:
“Survival differences across key oncogenic pathways varied by ancestry among pancreatic cancer patients. In the H/L cohort, WNT pathway alterations were associated with a survival probability of 0.00 at 71.7 months, compared to 0.39 in patients without alterations at 24.5 months (95% CI: 0.20–0.74). For TGF-Beta pathway alterations, survival was 0.30 (95% CI: 0.06–1.00) among altered cases versus 0.34 (95% CI: 0.14–0.81) in non-altered cases. TP53 alterations in H/L patients were associated with a 0.29 survival probability at 71.7 months (95% CI: 0.10–0.78), while non-altered patients had a survival of 0.50 at 7.4 months (95% CI: 0.13–1.00). For the PI3K pathway, survival for non-altered H/L patients was 0.30 at 71.7 months (95% CI: 0.11–0.81), while no long-term survival data were available for the altered group. Similarly, RTK/RAS pathway alterations yielded a survival of 0.27 at 71.7 months (95% CI: 0.10–0.73), with no non-altered group data at that timepoint. Among NHW patients, survival differences were also observed. WNT pathway alterations were linked to a 0.11 survival probability at 87.1 months (95% CI: 0.04–0.33), compared to 0.18 in non-altered patients at 80.9 months (95% CI: 0.14–0.23). For TGF-Beta, survival was 0.17 for both altered (69.9 months, 95% CI: 0.11–0.26) and non-altered groups (87.1 months, 95% CI: 0.13–0.23). TP53-altered patients had a survival of 0.16 at 87.1 months (95% CI: 0.12–0.21), whereas non-altered patients had 0.22 survival at 80.9 months (95% CI: 0.14–0.34). PI3K-altered patients had a survival of 0.09 at 62.8 months (95% CI: 0.02–0.49), while non-altered patients had 0.18 survival at 87.1 months (95% CI: 0.14–0.23). RTK/RAS pathway alterations were associated with a survival of 0.27 at 73.7 months (95% CI: 0.14–0.41), with no final timepoint data available for the non-altered group.”
The added paragraph of the Discussion section, lines 377 - 392, now read:
“Our findings highlight notable population-specific differences in pathway-associated survival among pancreatic cancer patients, with particularly stark contrasts observed in the WNT and TP53 pathways. Among H/L patients, WNT alterations were associated with complete loss of survival by the final timepoint, suggesting a potentially aggressive disease course in this group. Conversely, NHW patients with WNT alterations exhibited slightly longer survival, though still poor, underscoring the prognostic importance of this pathway across ancestries. TP53 alterations were also associated with reduced survival in both populations; however, H/L patients without alterations exhibited unexpectedly higher short-term survival, possibly reflecting differences in tumor biology or healthcare access. The PI3K and RTK/RAS pathways showed consistently poor outcomes when altered, with limited data on non-altered comparators, indicating the need for deeper investigation into these oncogenic drivers. Interestingly, TGF-Beta pathway survival was comparable between altered and non-altered groups in NHW patients but more variable among H/L patients. These findings suggest that the prognostic significance of specific pathways may differ by ancestry, reinforcing the importance of population-inclusive genomic profiling and survival analysis to guide precision oncology approaches.”
Comment 5:
“Language and Clarity: The manuscript contains long, overly technical sentences, redundant phrasing, and occasional grammatical errors. A professional English editing service is recommended to improve clarity and flow, particularly in the introduction and discussion”.
Response:
We thank Reviewer 2 for the helpful feedback regarding language and clarity. In response, we carefully re-wrote a substantial portion of the manuscript, with particular attention to the Introduction and Discussion sections, to improve readability, eliminate redundancy, and simplify overly technical language. These revisions were made to enhance the clarity and flow of the text while maintaining scientific precision. We believe the updated version is now more accessible to a broad scientific audience, and we appreciate the reviewer’s recommendation, which significantly strengthened the overall presentation of the manuscript.
Comment 6:
“Supplementary Material Transparency: Table S1 contains important annotation details that should be referenced more explicitly in the main text. Consider directly integrating key pathway mutation definitions into the Methods or Results for greater accessibility”.
Response:
We thank Reviewer 2 for this valuable suggestion regarding the transparency and accessibility of Supplementary Table S1. In response, we have revised the Methods and Results sections to more explicitly reference Table S1, highlighting its role in detailing gene-level mutation annotations for each pathway analyzed. Additionally, we have integrated key pathway mutation definitions directly into the Methods section to improve clarity and ensure that readers can easily understand how pathway alterations were defined without relying solely on the supplementary material. These updates enhance the transparency and reproducibility of our analysis, and we appreciate the reviewer’s input in helping us strengthen this aspect of the manuscript.
The added paragraph of the Methods section, lines 139-147, now read:
“To improve clarity and reproducibility, we defined each of the five oncogenic pathways—TGF-Beta, RTK/RAS, WNT, PI3K, and TP53—based on nonsynonymous mutations in key genes associated with each pathway. For example, TGF-Beta pathway alterations included mutations in SMAD2, SMAD3, SMAD4, TGFBR1/2, and ACVR2A/B; RTK/RAS included KRAS, HRAS, ALK, ERBB4, MET, and related genes; WNT included APC, CTNNB1, and AXIN1/2; PI3K included PIK3CA, PTEN, AKT1–3, and MTOR; and TP53 included TP53, CDKN2A, ATM, MDM2, and CHEK2. Gene-level frequencies are provided in Supplementary material, and these definitions were applied consistently across all analyses.”
Comments on the Quality of English Language:
“The manuscript is generally understandable, but the quality of English requires improvement. Several sections contain long, overly complex sentences, awkward phrasing, and redundant expressions. These issues affect clarity and may hinder reader comprehension. A thorough professional language edit is strongly recommended to improve sentence structure, grammar, and overall readability. Refining the language will enhance the manuscript’s impact and ensure the scientific content is communicated more effectively”.
Response:
We thank Reviewer 2 for the thoughtful and constructive feedback regarding language quality. In response, we have undertaken a thorough revision of the manuscript to improve clarity, grammar, and overall readability. We restructured long and complex sentences, removed redundant phrasing, and refined awkward language throughout the text—particularly in the Introduction, Results, and Discussion sections.
These revisions were made to ensure that the scientific content is communicated more clearly and effectively. We appreciate the reviewer’s suggestion, which has helped strengthen the overall quality and impact of the manuscript.

Round 2
Reviewer 1 Report
Comments and Suggestions for Authors
The authors do not possess basic pathological knowledge to publish an article like this. Pancreatic cancer means malignant tumors, than they list benign lesions, as well. PC abbreviation in itself is not benefitial to use. In the table that lists histological subtypes, most of them are not accurate, and are not in accordance with the current WHO classification. What is 'Pancreas' there? Pancreas adenocarcinoma means ductal? What is cystic tumor? That surely is no cancer. Mucinous neoplasms, osteoclastic giant cell tumor, and solid pseudopapillary neoplasm are also not cancers. In sample type, metastatic should be excluded completely, while there are no cases. This article is completely unacceptable without stage and grade, while these would affect the mutation profile. Table 2 is completely useless and unprofessional. Same applies to Table 3. The response letter should not just include the control-copied version of the modified section.
This project needs a pathologist to be involved.
Author Response
Responses to Reviewer 1's comments are provided in the attached Word document: "Response_Reviewer_1_Comments_050125_EV.docx"
Reviewer 1 Comments
This manuscript is resubmitted for your consideration with substantial revisions aimed at strengthening both its scientific rigor and clarity. We have carefully addressed all reviewer feedback and refined the manuscript to better articulate the significance of our findings to the broader scientific and clinical research communities.
In this study, we conducted a comprehensive pathway-level analysis of pancreatic cancer by leveraging one of the few publicly available genomic datasets with sufficient representation of Hispanic/Latino (H/L) patients. Our investigation represents a novel, ethnicity-focused approach to examining pathway-specific genomic alterations in pancreatic cancer, with emphasis on major oncogenic drivers including TGF-Beta, RTK/RAS, WNT, PI3K, and TP53.
Noteworthy among our findings is the identification of recurrent gene alterations in SMAD2, ERBB4, ALK, and CTNNB1 specifically among H/L patients. These results highlight the critical need for ethnicity-informed molecular profiling to potentially enhance risk assessment, facilitate early detection, and inform targeted treatment strategies in underserved populations. We are confident that our study provides a valuable contribution to the field of cancer health disparities—an area in urgent need of expanded research—and advances the growing efforts in precision oncology.
Thank you for taking the time to review our manuscript. Below, we provide detailed responses to the Round 2 reviewer comments. All corresponding revisions have been incorporated into the re-submitted Word document and are indicated in blue font and highlighted in yellow in the re-submitted Word file.
In accordance with Reviewer 1’s Round 2 feedback, we have addressed all comments from Round 1, as no additional concerns from the initial review were raised. We interpret this as a positive indication that the manuscript is now well-polished. Additionally, Reviewer 1 noted that “The English is fine and does not require any improvement,” which we appreciate.
Comment 1:
- “The authors do not possess basic pathological knowledge to publish an article like this”.
Response: Thank you for your comment. We respectfully disagree with this assessment. Our team comprises investigators from top 5 leading cancer research institutions in the United States, including NCI-designated comprehensive cancer centers. We have extensive expertise in the molecular characterization of gastrointestinal cancers, with a strong focus on cancer disparities. Our research has resulted in numerous peer-reviewed publications in scientific journals and has contributed meaningfully to advancing cancer equity on both national and global scales (PMIDs: 40282501, 40282485, 40227607, 40227587, 40165548, 40075719, 39999309).
In addition, our work is supported by several competitive research grants from the National Institutes of Health (NIH) and the National Cancer Institute (NCI), encompassing basic, translational, and clinical trial research—three of which are acknowledged in the funding section of this manuscript.
This study utilizes publicly available clinical/genomic datasets and does not involve histopathological evaluation or image-based tissue interpretation, which would require a pathologist. Therefore, the inclusion of a pathologist is not necessary for the current analysis. However, our team’s prior work includes multiple publications in spatial biology, spatial transcriptomics, spatial proteomics and single cell (PMIDs: 39711954, 32587328), which require and reflect a strong understanding of cancer pathology and spatial tissue architecture.
We remain confident that our training, institutional affiliations, and scientific contributions clearly demonstrate the qualifications necessary to conduct and publish this work.
Comment 2:
- “Pancreatic cancer means malignant tumors, than they list benign lesions, as well. PC abbreviation in itself is not benefitial to use. In the table that lists histological subtypes, most of them are not accurate, and are not in accordance with the current WHO classification. What is 'Pancreas' there? Pancreas adenocarcinoma means ductal? What is cystic tumor? That surely is no cancer. Mucinous neoplasms, osteoclastic giant cell tumor, and solid pseudopapillary neoplasm are also not cancers”.
Response: Thank you for this detailed feedback. We would like to clarify several points.
First, the acronym “PC” for pancreatic cancer is widely adopted in the literature and broadly understood by the scientific and clinical research communities (PMID: 40247202). We therefore believe its use is appropriate and beneficial for clarity and consistency throughout the manuscript.
Regarding tumor classification, we defined pancreatic cancer in accordance with the literature and clearly focused the background of our introduction on malignant pancreatic tumors, particularly pancreatic adenocarcinoma, which represents the majority of our analyzed cases due to sample size considerations.
The tumor subtype data used in our study were obtained from publicly available genomic datasets curated and annotated by expert contributors, such as those integrated into cBioPortal. These classifications reflect the established categories adopted by the research community and were not assigned by our team. In our analyses, we aimed to be inclusive of all tumor types classified as “pancreatic cancer” by these databases. This approach is consistent with our methodology in prior publications pointed in last comment, where we have addressed the inclusion of rare or histologically diverse subtypes, even when they are underrepresented, acknowledging their potential dilution in larger datasets.
We recognize the importance of accurate histological terminology and note that this issue was addressed in the manuscript text. While terms like “Pancreas,” “cystic tumor,” or others may appear in the database with varying granularity, they reflect the original annotations and not arbitrary labels from our group. In our current study, the primary emphasis remains on pancreatic adenocarcinoma, given its predominance in the dataset and its clinical relevance.
Finally, we respectfully note that the title of our manuscript—“Pathway-specific genomic alterations in pancreatic cancer across populations at risk”—accurately reflects the scope and inclusivity of the data used, while focusing the genomic analysis primarily on malignant entities.
Comment 3:
- “In sample type, metastatic should be excluded completely, while there are no cases”.
Response: Thank you for your observation. We respectfully believe that including the “metastatic” category in Table 1, even though it is represented as 0%, is important for transparency and methodological completeness. This reflects our data extraction strategy and ensures that all potential sample types considered in the analysis are clearly communicated. Including this category is particularly relevant for clinical readers, who routinely assess whether primary or metastatic samples were analyzed, as this distinction may influence the interpretation of genomic data. We have retained it intentionally to maintain clarity and reproducibility of our methodology.
Comment 4:
- “Table 2 is completely useless and unprofessional. Same applies to Table 3”.
Response:
We appreciate your feedback, though we respectfully disagree with this assessment. The structure and content of Table 1 in our manuscript closely align with many published "Table 1" formats in scientific journals such as those listed in the comment 1. Our classification of tumor types is based on curated data from cBioPortal, which is widely used and maintained by experts in the field. These classifications are standardized and represent the original annotations provided by the contributing institutions and tumor boards.
Tables 1 and 2 are integral to our manuscript and serve distinct and important purposes. Table 1 presents patient demographics and clinical characteristics for the Hispanic/Latino (H/L) and Non-Hispanic White (NHW) pancreatic cancer (PC) cohorts, offering readers crucial context regarding sample composition and disease subtypes. Clinicians and researchers alike often refer to such tables to understand cohort structure, which supports the interpretation of subsequent genomic findings. Including categories such as “metastatic” (even with 0% frequency) ensures methodological transparency and reflects common practice in clinical genomic studies.
Table 2 highlights ethnicity-associated differences in clinical and genomic features, including median mutation counts, tumor mutational burden (TMB), fraction of genome altered (FGA), histologic subtypes based on OncoTree codes, and key gene mutations. These comparisons are statistically tested and provide essential insights into population-specific molecular differences—core to the objectives of our study, which aims to identify pathway-specific alterations relevant to cancer disparities.
We respectfully believe these tables add significant value by presenting data clearly and reproducibly, and align with accepted scientific standards.
Comment 5:
- “The response letter should not just include the control-copied version of the modified section”.
Response: Thank you for bringing this to our attention. In our initial response, we included the modified sections directly in the letter—along with page and line numbers—out of consideration for your time and with the intention of making the review process more efficient and transparent. Our goal was to provide clear access to all changes to support a comprehensive evaluation.
However, as per your request in this second round, we have omitted the control-copied text and now provide concise responses with precise references.
Comment 6:
“This project needs a pathologist to be involved”.
Response: Thank you for your comment. As discussed in our response to a previous comment, we respectfully disagree with this assessment, as this study is based entirely on publicly available clinical and genomic datasets that have already been curated and annotated by expert contributors. Our analysis does not involve histopathological image review, tissue diagnosis, or cell-level classification—areas where a pathologist’s expertise would be essential. Given the nature of the data and the objectives of our pathway-level genomic analysis, the involvement of a pathologist is not required for the scope of this work.
We sincerely appreciate your time and effort in reviewing our work.

Reviewer 2 Report
Comments and Suggestions for Authors
The revised manuscript demonstrates significant improvement in both clarity and scientific framing. The rationale for focusing on pathway-level mutations in pancreatic cancer among different ethnic groups, particularly Hispanic/Latino populations, is now well justified in the introduction. The discussion also more appropriately contextualizes the findings within the current landscape of cancer disparity research.
Adding confidence intervals and improved survival figure annotations strengthens the statistical interpretation. Pathway definitions are now clearly explained in the main text and linked to Supplementary Table S1, which improves methodological transparency.
Importantly, the lack of clinical covariates such as tumor stage and treatment information is now acknowledged as a limitation, and the discussion offers reasonable caution regarding the interpretation of survival differences.
That said, a few areas still require attention. First, while you have moderated the conclusions appropriately, some language still implies potential clinical applicability; we recommend framing such statements as exploratory or hypothesis-generating. Second, although the writing is much improved, a final round of professional English editing is recommended to correct minor grammatical issues and enhance sentence flow.
Lastly, the text in several figures (particularly figure labels and axis text) remains too small to read comfortably. We recommend increasing the font size across all figures and ensuring clarity and resolution before final submission.
The revised manuscript's English language has improved, with better sentence structure and clearer organization overall. However, several sections still contain overly long or complex sentences, minor grammatical errors, and occasional redundancies that affect readability. A final round of professional language editing is recommended to improve clarity, consistency, and overall flow, particularly in the introduction and discussion sections. This will help ensure the scientific message is communicated more effectively to a broad audience.
Author Response
Responses to Reviewer 2's comments are provided in the attached Word document: "Response_Reviewer_2_Comments_050225_EV.docx".
Reviewer 2 Comments
This manuscript is resubmitted for your consideration with substantial revisions aimed at strengthening both its scientific rigor and clarity. We have carefully addressed all reviewer feedback and refined the manuscript to better articulate the significance of our findings to the broader scientific and clinical research communities.
In this study, we conducted a comprehensive pathway-level analysis of pancreatic cancer by leveraging one of the few publicly available genomic datasets with sufficient representation of Hispanic/Latino (H/L) patients. Our investigation represents a novel, ethnicity-focused approach to examining pathway-specific genomic alterations in pancreatic cancer, with emphasis on major oncogenic drivers including TGF-Beta, RTK/RAS, WNT, PI3K, and TP53.
Noteworthy among our findings is the identification of recurrent gene alterations in SMAD2, ERBB4, ALK, and CTNNB1 specifically among H/L patients. These results highlight the critical need for ethnicity-informed molecular profiling to potentially enhance risk assessment, facilitate early detection, and inform targeted treatment strategies in underserved populations. We are confident that our study provides a valuable contribution to the field of cancer health disparities—an area in urgent need of expanded research—and advances the growing efforts in precision oncology.
Thank you for taking the time to review our manuscript. Below, we provide detailed responses to the Round 2 reviewer comments. All corresponding revisions have been incorporated into the re-submitted Word document and are indicated in blue font and highlighted in yellow in the re-submitted Word file.
Reviewer 2 provided positive and constructive feedback on the revised manuscript. The reviewer acknowledged that the manuscript now demonstrates significant improvement in both clarity and scientific framing, particularly in justifying the focus on pathway-level mutations in pancreatic cancer across ethnic groups, with emphasis on Hispanic/Latino populations. They noted that the rationale is now well articulated in the Introduction, and that the Discussion appropriately contextualizes the findings within the broader field of cancer disparity research.
Reviewer 2 also commended several key improvements: the addition of confidence intervals and enhanced survival figure annotations, which together improve the statistical interpretation of our results; the clearer explanation of pathway definitions within the main text and improved linkage to Supplementary Table S1, which enhances methodological transparency; and the acknowledgment of limitations, including the absence of clinical covariates such as tumor stage and treatment, along with appropriately tempered interpretation of survival differences.
While the reviewer noted a few remaining areas for refinement, we sincerely appreciate their thoughtful critique. Their feedback has been instrumental in strengthening the manuscript’s clarity, rigor, and translational relevance.
Reviewer 2 writes:
“The revised manuscript demonstrates significant improvement in both clarity and scientific framing. The rationale for focusing on pathway-level mutations in pancreatic cancer among different ethnic groups, particularly Hispanic/Latino populations, is now well justified in the introduction. The discussion also more appropriately contextualizes the findings within the current landscape of cancer disparity research.
Adding confidence intervals and improved survival figure annotations strengthens the statistical interpretation. Pathway definitions are now clearly explained in the main text and linked to Supplementary Table S1, which improves methodological transparency.
Importantly, the lack of clinical covariates such as tumor stage and treatment information is now acknowledged as a limitation, and the discussion offers reasonable caution regarding the interpretation of survival differences. That said, a few areas still require attention.”
We appreciate Reviewer 2’s thoughtful and supportive comments. It is encouraging to know that the reviewer highlighted the significance of our study, particularly its focus on ethnicity-specific genomic alterations in pancreatic cancer using large-scale, publicly available datasets. We are grateful for the recognition that our pathway-level mutation analysis and its connection to clinical outcomes addresses a critical gap in cancer disparities research. In response to the reviewer’s suggestions, we have made targeted revisions to better articulate the study’s novelty, enhance the interpretation of key findings, and improve the overall clarity and impact of the manuscript.
Comment 1:
- “First, while you have moderated the conclusions appropriately, some language still implies potential clinical applicability; we recommend framing such statements as exploratory or hypothesis-generating”.
Response:
Thank you for this important and thoughtful observation. We agree that, given the retrospective and observational nature of our study, statements regarding clinical implications should be presented with appropriate caution. In response, we have further revised the Conclusions and relevant sections of the Discussion to clearly frame our findings as exploratory and hypothesis-generating. This adjustment ensures that the interpretation remains consistent with the study’s design and reinforces the need for future validation in prospective and clinically annotated cohorts.
For new changes, the latest adjustments are listed below:
The revised paragraph of the Simple Summary section, lines 17-20, now reads as follows:
“These findings offer exploratory insights into potential molecular differences in pancreatic cancer across ethnic groups. While preliminary, these observations may inform future hypothesis-driven research aimed at understanding how genomic variability could contribute to disparities in patient outcomes and therapeutic responses.”
The revised paragraph of the Materials and Methods section, lines 126-130, now reads as follows:
“This approach enabled us to explore whether certain molecular alterations appeared more frequently in specific racial/ethnic groups, offering preliminary insights into genomic heterogeneity that may help generate hypotheses about differential treatment responses.”
The revised paragraph of the Results section, lines 198-201, now reads as follows:
“These findings highlight exploratory patterns of ethnicity-associated molecular variation in pancreatic cancer, suggesting that Hispanic/Latino patients may exhibit distinct genomic profiles. While preliminary, these observations could help inform future studies on tumor progression and therapeutic response.”
The revised paragraph of the Results section, lines 350-356, now reads as follows:
“However, the lack of statistical significance in most comparisons underscores the need for larger, well-powered studies to validate these preliminary observations. Further research is necessary to explore their potential relevance to precision medicine and targeted therapy in Hispanic/Latino pancreatic cancer patients. Understanding these molecular patterns may ultimately contribute to hypotheses about ethnicity-specific tumor biology and help inform future investigations aimed at improving outcomes in underrepresented populations.”
The revised paragraph in the Discussion section, lines 412–419, now reads as follows:
“Our findings suggest that pathway-specific genomic profiling may offer preliminary insights into pancreatic cancer risk and potential therapeutic targets across diverse populations. The observed higher frequency of SMAD2, ERBB4, ALK, and CTNNB1 mutations in Hispanic/Latino patients may warrant further investigation as potential biomarkers for early detection. Similarly, the differing patterns of SMAD2 mutations in H/L patients and SMAD4 mutations in Non-Hispanic White patients could reflect underlying biological differences, though additional studies are needed to explore these ethnicity-associated mechanisms and their clinical relevance.”
The revised paragraph in the Discussion section, lines 462–465, now reads as follows:
“These observations support the hypothesis-generating value of investigating age- and ethnicity-associated molecular drivers, which may guide future research into early detection and the development of more tailored treatment approaches.”
The revised paragraph in the Conclusions section, lines 511-514, now reads as follows:
“These findings may serve as a basis for future hypothesis-generating studies focused on ethnicity-specific molecular profiling, with the goal of exploring potential avenues for risk stratification, early detection, and targeted treatment approaches.”
Comment 2:
“Second, although the writing is much improved, a final round of professional English editing is recommended to correct minor grammatical issues and enhance sentence flow”.
Response:
Thank you for your thoughtful recommendation. We appreciate your recognition of the improved clarity in this revised version. In response, we conducted an additional comprehensive round of professional English editing focused on refining sentence structure, improving transitions between ideas, and correcting minor grammatical issues. We carefully reviewed each section of the manuscript to ensure smoother flow and enhanced readability, with the goal of presenting our findings as clearly and effectively as possible. We believe these refinements have further strengthened the quality and presentation of the manuscript.
For instance, in the introduction, the transition between paragraphs was improved for better flow.
The revised paragraphs in the Introduction section, lines 71-96, now read as follows:
“The TGF-Beta signaling pathway has a dual role in PC, acting as a tumor suppressor in early stages but promoting epithelial-to-mesenchymal transition (EMT), invasion, and metastasis in advanced disease [16]. In PC, mutations in SMAD4 and TGFBR2 are commonly observed and have been implicated in disease progression and therapy re-sistance [17].
Building on this, the RTK/RAS pathway plays a central role in PC pathogenesis, with KRAS mutations occurring in over 90% of cases, driving uncontrolled cell growth and conferring resistance to targeted therapies [18,19]. Mutations in additional RTK/RAS pathway genes, including ERBB4, ALK, and HRAS, may further contribute to tumor progression and therapeutic resistance [20,21].
In parallel, the PI3K/AKT pathway is a critical regulator of tumor metabolism and sur-vival, with frequent alterations in PTEN and PIK3CA driving increased tumor inva-siveness and immune evasion [22]. Mutations in PI3K pathway genes have been associated with poor prognosis and treatment resistance, making them important targets for preci-sion medicine approaches [23-25].
Equally critical, the TP53 pathway plays a fundamental role in genomic stability, apoptosis, and cell cycle regulation. TP53 mutations are among the most common alter-ations in PC, and emerging evidence suggests that mutant p53 interacts with KRAS to drive metastasis [26,27]. In addition to TP53 mutations, CDKN2A deletions are frequently observed in PC and are associated with aggressive tumor behavior [28,29].
Finally, the WNT/β-catenin signaling pathway is another key driver of PC progression, regulating cell proliferation and differentiation. WNT pathway mutations have been identified in nearly all PC cases, with alterations in genes such as CTNNB1 and RNF43 contributing to tumor initiation and progression [30]. Aberrant WNT signaling has also been linked to chemoresistance, underscoring the need for pathway-specific therapeutic interventions [31].”
Comment 3:
Lastly, the text in several figures (particularly figure labels and axis text) remains too small to read comfortably. We recommend increasing the font size across all figures and ensuring clarity and resolution before final submission”.
Response:
Thank you for this helpful comment. We agree that figure readability is essential for effective data presentation. In response, we have updated all our figures (Figure 1 and Supplementary Figure S1) by increasing the font size of all labels and axis text and improving overall resolution to ensure clarity. We appreciate your attention to this detail and believe these changes enhance the visual quality of the figures.
We sincerely thank the reviewer for taking the time to carefully evaluate our manuscript and provide thoughtful, constructive feedback. Your detailed comments and recommendations have been instrumental in guiding meaningful revisions that have strengthened the clarity, scientific rigor, and overall quality of our work. We truly appreciate your commitment to improving the manuscript and your contributions to the peer review process.
